# The stability of multitrophic communities under habitat loss

Chris McWilliams[1], Miguel Lurgi [2], Jose M. Montoya [2], Alix Sauve[3] & Daniel Montoya [2]

Habitat loss (HL) affects species and their interactions, ultimately altering community dynamics. Yet, a challenge for community ecology is to understand how communities with multiple interaction types—hybrid communities—respond to HL prior to species extinctions. To this end, we develop a model to investigate the response of hybrid terrestrial communities to two types of HL: random and contiguous. Our model reveals changes in stability—temporal variability in population abundances—that are dependent on the spatial configuration of HL. Our findings highlight that habitat area determines the variability of populations via changes in the distribution of species interaction strengths. The divergent responses of communities to random and contiguous HL result from different constraints imposed on individuals' mobility, impacting diversity and network structure in the random case, and destabilising communities by increasing interaction strength in the contiguous case. Analysis of intermediate HL suggests a gradual transition between the two extreme cases.

[1] School of Computer Science, Electrical and Electronic Engineering, and Engineering Maths, Merchant Venturers Building, University of Bristol, Bristol BS81UB, UK. [2] Centre for Biodiversity Theory and Modelling, Theoretical and Experimental Ecology Station, CNRS, 2 route du CNRS, 09200 Moulis, France. [3] University of Bordeaux, Integrative and Theoretical Ecology LabEx COTE, 33615 Pessac, France. Correspondence and requests for materials should be addressed to D.M. (email: daniel.montoya@sete.cnrs.fr)

Habitat loss (hereafter, HL) is one of the main threats to biodiversity, driving both local and global extinctions of species[1]. In his 'empty forest' hypothesis, Janzen first remarked that, in addition to species extinctions, HL leads to the 'most insidious type of extinction': the extinction of ecological interactions[2]. This has also been referred to as the 'missed component of biodiversity loss' that accompanies or precedes species extinctions[3]. This observation prompted research to understanding the effects of HL on communities with particular emphasis on species-interaction networks[3–7]. These studies show that mutualistic and trophic networks respond differently to HL: whereas mutualistic networks tend to break into smaller sub-networks[6] with lower nestedness (the degree to which the diets of consumers are proper subsets of other, more generalist consumers) and reduced interaction strengths[8], trophic networks lose modularity (the extent to which interactions occur more frequently within modules than between modules) and increase in interaction strength[5]. In all cases, both types of networks shift towards structures that have shown to beget unstable community dynamics[9,10].

Such network changes are not a mere by-product of species extinction caused by HL. For instance, host-parasitoid communities subjected to a gradient of habitat modification show structural changes in their interaction network, i.e. lower evenness of interaction frequencies and ratio of parasitoid to host species, without any significant modification in species richness[11]. Similarly, empirical data on insect food-webs show that interaction diversity declines faster than species diversity with HL[12]. These observations raise an important question: do changes in network structure precede species extinctions?

HL is a complex phenomenon. It entails the fraction of destroyed habitat, but also several features of how the loss takes place. Whether destroyed habitat patches are spatially-correlated (contiguous HL) or are randomly distributed throughout the landscape (random HL) is important, as this can influence the mobility of individuals across the landscape and, in turn, the encounter probability between potentially-interacting species. Thus, the effects of HL on species interactions may depend on the spatial patterning of HL, i.e. random vs contiguous. A number of modelling studies have concluded that extinction thresholds—the point at which species richness experiences an abrupt decline following a disturbance—occur at higher fractions of HL when HL is contiguous, as opposed to random loss[13,14], because spatially-correlated loss leaves larger areas of pristine habitat where species can persist[13]. The differential effects of the type of HL are also supported by empirical findings[5,15]. Therefore, the effect of HL on interaction frequencies and thus on the stability of multitrophic communities is likely to depend on the nature of HL.

Two major gaps exist in our understanding of how HL affects ecosystems. Firstly, multiple types of interaction (e.g. trophic and mutualistic) are rarely considered simultaneously[16–21]. Hence, we know little about HL-induced changes that precede species extinctions in communities with different architectures and interactions types. Secondly, ecologists tend to focus on species persistence, while HL is likely to affect other aspects of community stability such as resistance, robustness or resilience[22,23]. Since habitat alterations can modify the structure of interaction networks[6,11,12], and network structure can in turn affect its stability[10,24,25], we can expect that HL impacts the stability of biological communities in the absence of species extinctions. Wang and Loreau[26] recently proposed a stability analogue to the species–area relationship, suggesting that temporal variability of population abundances scales inversely with habitat area[27]. Also, empirical[4] and theoretical[25] studies suggest that variability may increase in smaller fragments of habitat. Different mechanisms have been suggested to explain such changes in temporal variability at different spatial scales, including spatial averaging over asynchronous response of species to environmental fluctuations[26], higher predation pressure[25], and an increase in interaction strengths[4]. Of these, only the latter has received empirical support[28], although in a context unrelated to HL. Despite evidence on bivariate relationships between area and stability, and between interaction strength and stability, it is not clear how HL can affect the relationship between interaction patterns and stability of multitrophic communities.

In this study, we conduct a systematic exploration of the responses of hybrid communities (communities with both trophic and mutualistic interactions) to different types of HL. We use an individual-based model of community dynamics driven by bio-energetics, spatial constraints, and species interactions. To investigate community responses to HL beyond its effects on species richness, we keep the number of species constant. We specifically address three key questions:

Firstly, how does the stability of hybrid communities respond to HL, and what mechanisms drive this response? We simultaneously investigate changes in network structure and temporal variability in population abundances (CV population) and the variability in species' range areas (CV range)[29]—over a gradient of HL. On one hand, we opt for CV population because there is evidence that interaction strengths can be altered by HL[5,8], which we hypothesise to be associated with changes in variability. On the other hand, HL can affect the mobility of individuals, which may in turn affect the spatial range of species and, consequently, the strength of interactions between species. Based on empirical evidence[28], we hypothesize that the impacts of HL on stability are mediated by changes in the distribution of interaction strengths.

Secondly, do changes in stability differ between different types of HL? We consider two types of HL (i.e. random and contiguous) because they represent two extremes of spatial correlation in the pattern of HL used in previous studies[30–37], and correspond to the two extreme cases of optimising land use (i.e. land-sparing vs land-sharing[38]). While under land-sparing biodiversity is essentially concentrated into one or a few large habitat fragments (i.e. the loss of habitat is contiguous), under land-sharing it is distributed across the whole landscape but in a large number of smaller, fragmented, patches of habitat—result of random HL. To provide insight into the transition between the two extreme cases, we further simulate an intermediate scenario of HL halfway between random and contiguous loss. We hypothesize that random HL will result in more spatially-fragmented communities compared with contiguous HL, and that communities experiencing random HL will show stronger negative effects on community structure and stability.

Thirdly, does the proportion of mutualistic to trophic interactions influence community responses to HL? Recent studies show that mutualistic interactions can either stabilize[21,39] or destabilize[40] the temporal dynamics of undisturbed hybrid communities; thus, although the sign of the effect is not clear, we expect that the fraction of mutualism will affect how stability responds under HL.

Our analysis indicates that changes in the structure and dynamics of ecological communities are strongly dependent on the spatial configuration of HL. Community responses to random and contiguous HL emerge from different constraints imposed on individual mobility patterns. Such differences impact diversity and network structure in the random case, and destabilise communities by increasing interaction strength in the contiguous case. These effects are not associated with changes in network structure, and are not qualitatively affected by the fraction of mutualism. The response of communities under intermediate HL lies in-between the random and contiguous HL, thus suggesting a gradual transition between the two extreme cases.

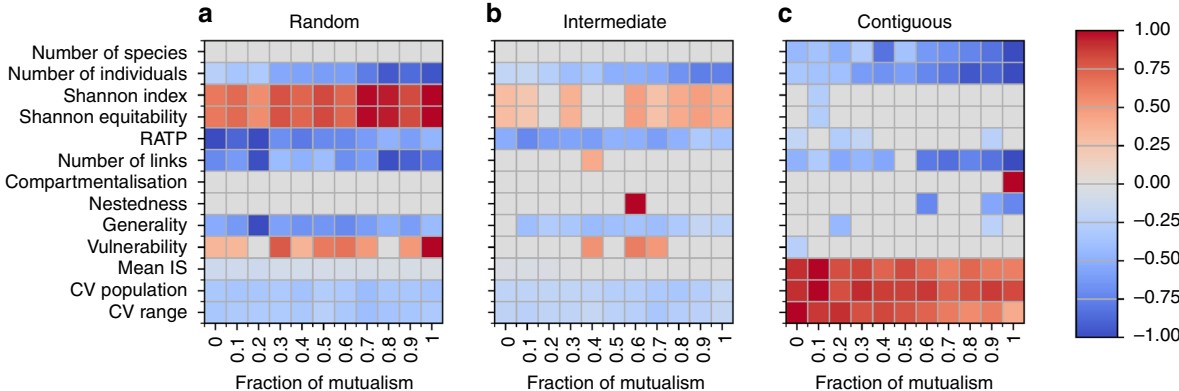

**Fig. 1** Response of community metrics to habitat loss. Summary of significant linear trends in the response variables under **a** random, **b** intermediate, and **c** contiguous HL, are shown for different fractions of mutualism. Red and blue squares represent significant (at $p \leq 0.05$; F-test) positive and negative linear trends, respectively, while grey indicates the absence of a significant trend. The shade of the red/blue indicates the size of trend detected, as shown in the colour bar. Trend size is normalized within each row, such that the colour indicates the effect size for each response variable relative to different fractions of mutualism (normalization was performed by setting the maximum value of each row to 1). This should be noted for the number of species, which, despite it decreases significantly under contiguous HL, the largest effect size detected corresponds to an average loss of ~0.5 species across the HL gradient. Therefore, all species can be considered to persist in the simulated communities

## Results

**Diversity and network structure**. Changes in diversity and network structure differed between random and contiguous HL (Fig. 1). Under random HL, abundances became more evenly distributed between species while the relative abundance of top predators (RATP) decreased. Prey species became more vulnerable, while predator species became more generalist in their diets. Intermediate HL produced similar responses for some of these metrics (e.g. distribution of abundances, RATP), yet the size of the trend was generally weaker. However, changes in network structure (i.e. species vulnerability and generality) were not observed under contiguous loss of habitat. The contrasted responses of network properties to random and contiguous HL scenarios were independent of changes in the number of links, which always decreased (due to reduced co-occurrence between interaction partners), suggesting that the way links were lost from the networks differed between HL scenarios. For some of these properties (e.g. RATP), changes observed across the gradient of HL were significant, however, the effect sizes were small (Supplementary Fig. 10). A list and definition of the variables used to describe the communities are provided in Table 1.

Interestingly, our results show that community responses across the three HL scenarios were gradual, going from random through intermediate and ending in contiguous loss (Fig. 2). This suggests that the magnitude of changes experienced by communities due to HL are predictable based on the degree of spatial autocorrelation of the lost habitat. Redundancy analysis (RDA) revealed that 20.26% of the variability across replicates can be explained by HL type. Furthermore, this RDA axis ordered community responses to HL across a continuum from random to contiguous loss (Fig. 2), which is consistent with a gradual transition.

**Stability**. Variability of population abundances grew with the fraction of habitat lost under the contiguous loss scenario. This response was strongly correlated with an increase in mean interaction strength (hereafter IS) (Fig. 3c; $R^2_{CV} = 0.54$, $R^2_{IS} = 0.64$). Conversely, random HL produced the opposite response: a decrease in temporal variability which correlated with reduced IS (Fig. 3a; $R^2_{CV} = 0.47$, $R^2_{IS} = 0.30$). We also observed a clear clustering of data points in IS—CV population space according to the amount of HL. Points describing community responses to small levels of random HL (40% loss or less) clustered towards

high IS and high CV population values, whereas we observed the opposite for the contiguous loss scenario (Fig. 3, panels D and F). This clustering exemplifies the contrasting effects of the different HL types on variability and IS. Changes in the variability of population abundances and IS for intermediate HL were in-between those observed for the random and contiguous scenarios (Fig. 3b; $R^2_{CV} = 0.33$, $R^2_{IS} = 0.007$), and the above clustering was not clearly observed (Fig. 3b, e). On the other hand, the mean coefficient of temporal variation in species range area (CV range) decreased under random and intermediate HL but increased under contiguous HL, suggesting various effects of fragmentation on individual mobility and species range area (Fig. 1).

**The role of mutualism**. The fraction of mutualism did not qualitatively alter the response of communities to HL (Fig. 1). All variables responded consistently to HL for at least 9 out of the 11 fractions of mutualism studied. The only two exceptions were nestedness and RATP, for which significant trends in changes occurred only at two fractions of mutualism. However, in these cases the effect sizes were small (Supplementary Figs. 9, 10).

In certain cases, the fraction of mutualism quantitatively affected community response by altering the magnitude of the effect (see colour scale in Fig. 1). In most cases this could be attributed to inherent differences between communities prior to HL (i.e. at HL = 0). Typically, communities with higher fractions of mutualism had larger abundances in pristine landscape, but this difference diminished across the HL gradient as all communities tended towards extinction (Supplementary Fig. 8). Similar reductions in the effect size across the HL gradient were observed for RATP (Supplementary Fig. 10).

**Routes to instability**. The net effect of HL on stability was opposite between HL types, with random loss causing a decrease (standardised model coefficient: −0.279) and contiguous loss an increase (0.312) in the temporal variability of populations. This is because many of the hypothesized relationships between diversity, network structure and stability (Supplementary Note 2) differed in sign or strength between random and contiguous HL, according to our linear models (Fig. 4, Table 2). Both random and contiguous HL directly decreased CV population (−0.111 and −0.080, respectively; Table 2). However, the total impact of HL on CV population was mainly mediated through indirect effects

**Table 1 List of variables to describe communities**

| Metric | | Definition |
|---|---|---|
| Diversity | Number of species: with at least one individual present in the landscape. | $S$ |
| | Number of individuals: | $N = \sum_{i=1}^{S} n_i$, where $n_i$ is the number of individuals belonging to species $i$. |
| | Shannon index: measures evenness in distribution of species abundances. | $D_{Shannon} = -\sum_{i=1}^{S} r_i log(r_i)$, where $r_i$ is the relative abundance of species $i$. |
| | Shannon equitability: is the Shannon index, normalised to control for number of species present. | $E_{Shannon} = \frac{D_{Shannon}}{log(S)}$, where $D_{Shannon}$ is the Shannon index defined above. |
| | RATP: relative abundance of top predator species. | $RATP = \left(\sum_{j=1}^{P} n_j\right) / \left(\sum_{i=1}^{S} n_i\right)$, where $P$ is the number of top predator species, $i$ indexes all species, and $j$ indexes top predators only. |
| Network | Number of links: the number of links present in the realised interaction network. (Presence defined as at least one interaction event in the landscape during 200 time steps.) | $L$ |
| | Compartmentalisation: the degree to which species share common neighbours across the network[51]. | $C = \frac{1}{S(S-1)} \sum_{j \neq i} \sum_{i=1}^{S} c_{ij}$, where $c_{ij}$ is the number of species with which both $i$ and $j$ interact, divided by the number of species with which neither $i$ or $j$ interact. |
| | Nestedness: the extent to which specialist species interact with subsets of the species with whom generalists interact[40]. | Calculated for the mutualistic sub-network only, using the NODF algorithm[52]. |
| | Generality: weighted quantitative generality[53]. | $G_q = \sum_{k=1}^{S} \frac{b_{.k}}{b_{..}} n_{N,k}$, where $b_{.k}$ is the total amount of biomass going into species $k$, and $b_{..}$ is the total amount of biomass flowing through the entire ecological network. $n_{N,k}$ is the number of species predated on by species $k$. Here, the biomass flowing from one species to another was calculated as the number of individuals of a given prey species eaten by individuals of predator species $k$. |
| | Vulnerability: weighted quantitative vulnerability[53]. | $V_q = \sum_{k=1}^{S} \frac{b_{k.}}{b_{..}} n_{P,k}$, where $b_{k.}$ is the total biomass emanating from species $k$. $b_{..}$ is the total amount of biomass flowing through the entire ecological network. $n_{P,k}$ is the number of predator species that feed upon prey species $k$. Here, the biomass flowing from one species to another was calculated as the number of individuals of prey species $k$ eaten by a given predator species. |
| | Mean interaction strength (IS): average inter-specific interaction strength (averaged over all interactions in realized network) | $\sum_{i \neq j} \frac{b_{ij}}{2n_i n_j}$, where $b_{ij}$ is the total biomass flowing from prey species $i$ to predator species $j$—quantified here as the total number of individuals (or fractions of it, in the case of plants) from species $i$ eaten by individuals of species $j$. This way of calculating interaction strengths quantifies the per-capita effect of a predator species over its prey, and it is thus analogous to Paine's index and Lotka–Volterra interaction coefficients[9,54]. Hence, these values allow to assess and understand community stability based on the strengths of ecological interactions. |
| Stability | CV population: mean coefficient of temporal variation in species population abundances. | $\frac{1}{S} \sum_{i=1}^{S} \frac{\sigma(n_i)}{\mu(n_i)}$, where $\mu(n_i)$ and $\sigma(n_i)$ are the mean and standard deviation in the abundance $n_i$ of species $i$ over 200 simulation time steps. |
| | CV range: mean coefficient of temporal variation in species range area. | $\frac{1}{S} \sum_{i=1}^{S} \frac{\sigma(a_i)}{\mu(a_i)}$, where $\mu(a_i)$ and $\sigma(a_i)$ are the mean and standard deviation in the range area $a_i$ of species $i$ over 200 simulation time steps. The range area of a species is defined as the area of the circle, centred on the centre of mass of the species spatial distribution, that contains 95% of the individuals belonging to that species. |

that differed in sign between the contiguous and the random HL (0.392 and −0.169, respectively; Table 2). This accounts for the marked differences between HL scenarios observed in Fig. 1, and reveals different major pathways through which HL affects stability of multitrophic communities (Fig. 4, Table 2). Intermediate HL directly decreased CV population (−0.086), and its total effect on stability was negative (−0.161).

Structural equation models (SEMs) therefore support our expectation that the impacts of HL on the stability of multitrophic communities are driven by changes in the distribution of interaction strengths. We identified four variables (Links, RATP, IS and CV range) mediating the effect of HL on stability. All four were always net contributors to CV population (Table 2), although the number of links had a much more pronounced effect in the contiguous HL scenario than in the random one. Under random HL (Fig. 4a), the strongest influence of HL on CV population was mostly mediated by CV range (0.308).

Conversely, HL affected CV population mostly through IS alone under contiguous HL (0.834, Fig. 4c), with a small negative contribution to this effect mediated through CV range (−0.045). The distribution of IS shifted to lower values under random HL, whereas they became stronger under contiguous HL (Supplementary Fig. 12). For intermediate HL, community response patterns are closer to random than to contiguous loss, but the standardised coefficients are significantly reduced (Fig. 4b).

**Mobility patterns under habitat loss**. Individuals explored a larger portion of the landscape under contiguous HL as compared with random or intermediate HL (Fig. 5). This was because of the contrasting effect of HL types on the spatial configuration of the remaining habitat: whereas contiguous loss compresses communities to smaller regions of habitat (Fig. 5d), random loss spatially segregates these communities by exposing them to fragmentation

(Fig. 5b). Intermediate HL is halfway between these two extreme scenarios (Fig. 5c), and this affects individual mobility patterns accordingly (Fig. 5e). This difference in individual mobility is associated with opposite changes in RATP, IS, CV range and CV

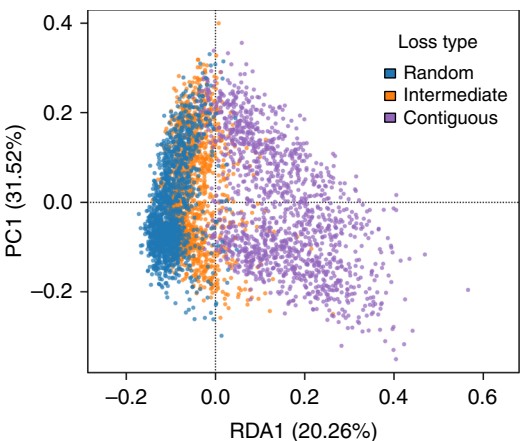

**Fig. 2** Community responses to habitat loss are determined by the type of loss. Redundancy analysis of community responses to HL constrained by HL type shows a clear separation of community responses to HL according to the type of loss experienced: random, intermediate or contiguous. The constrained (RDA1) and unconstrained (PCA1) axes (RDA1) explain a 20.26% and 31.52%, respectively, of the variability observed in the data

population (Figs. 1 and 4), suggesting that mobility patterns influence those metrics.

## Discussion

HL has major consequences for the stability of communities within the remaining areas of habitat. Our model of a spatial ecological network of trophic and mutualistic interactions suggests that: (1) the type of HL strongly mediates changes in stability in terms of temporal variability of population abundances, (2) changes in stability are largely driven by changes in the distribution of interaction strengths following area loss, and (3) changes in stability are independent of the number of species and the structural properties of the network of interactions. Our analysis of three scenarios of HL suggests that community responses are approximately gradual and predictable based on degree of spatial autocorrelation of the lost habitat. In particular, our results highlight a strong relationship between habitat area, IS, and variability of multitrophic communities, which is consistent with empirical studies linking strong interactions and high variability in population dynamics[9,28]. It also suggests that the reported relationships between habitat area, IS and stability are generalizable to terrestrial communities of multiple species and trophic and mutualistic interactions.

Changes in stability—the temporal variability in population abundances—of multitrophic communities following HL are associated with changes in a limited number of community properties (Table 2, Fig. 1). However, the effects of HL on these properties depend on the type of HL. Contiguous loss compresses communities to smaller fractions of habitat where population

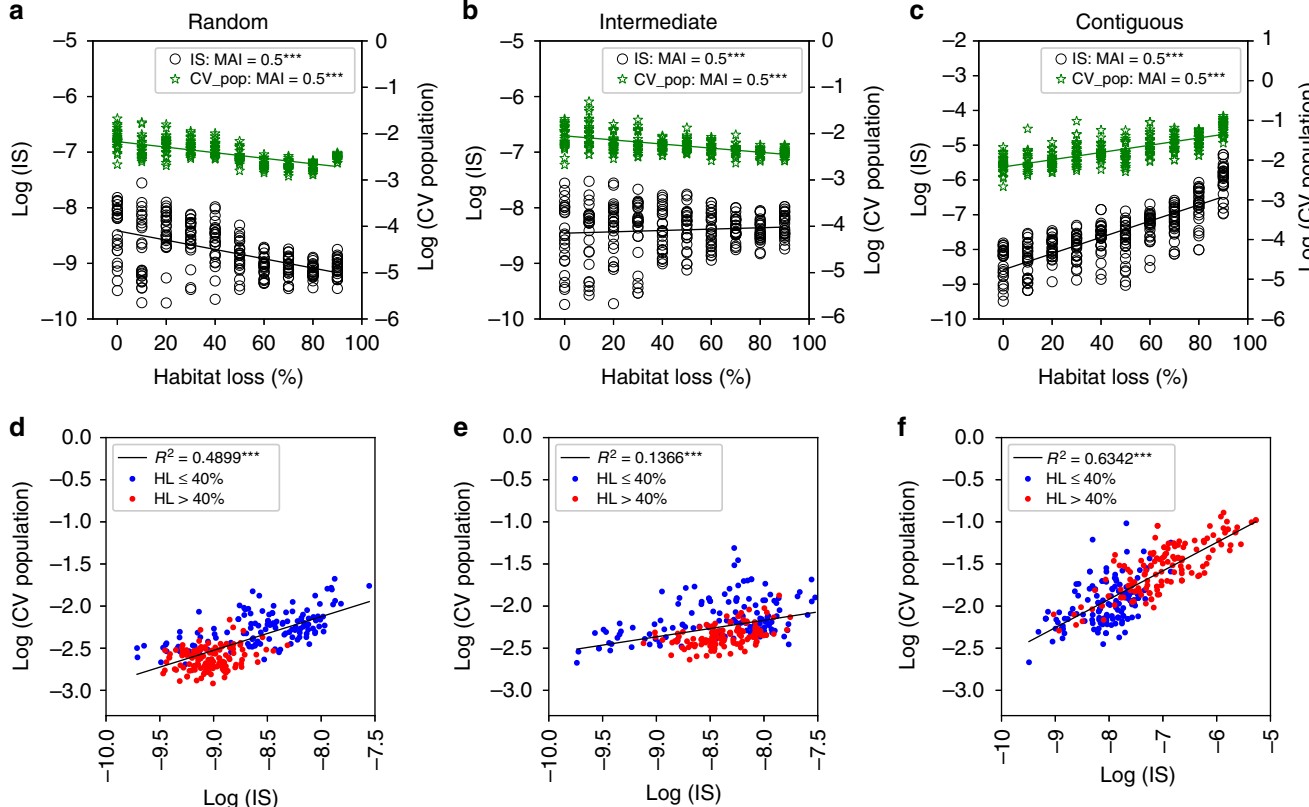

**Fig. 3** Habitat loss, interaction strengths and temporal variability. Interactions strengths and temporal variability are natural log-transformed to linearize trends. **a–c** Mean interaction strength (IS) averaged over all interactions in realized network, and mean coefficient of variation in species abundances averaged over all species (CV population). **d–f** IS as a linear predictor for mean CV population, with low and high HL communities indicated by blue and red circle respectively. All communities for fractions of mutualism equal to 0.0, 0.5, 1.0 are shown (***$p$-value < 0.05; F-test). **a** $R^2_{CV} = 0.30$, $R^2_{IS} = 0.47$; **b** $R^2_{CV} = 0.33$, $R^2_{IS} = 0.007$; **c** $R^2_{CV} = 0.54$, $R^2_{IS} = 0.64$

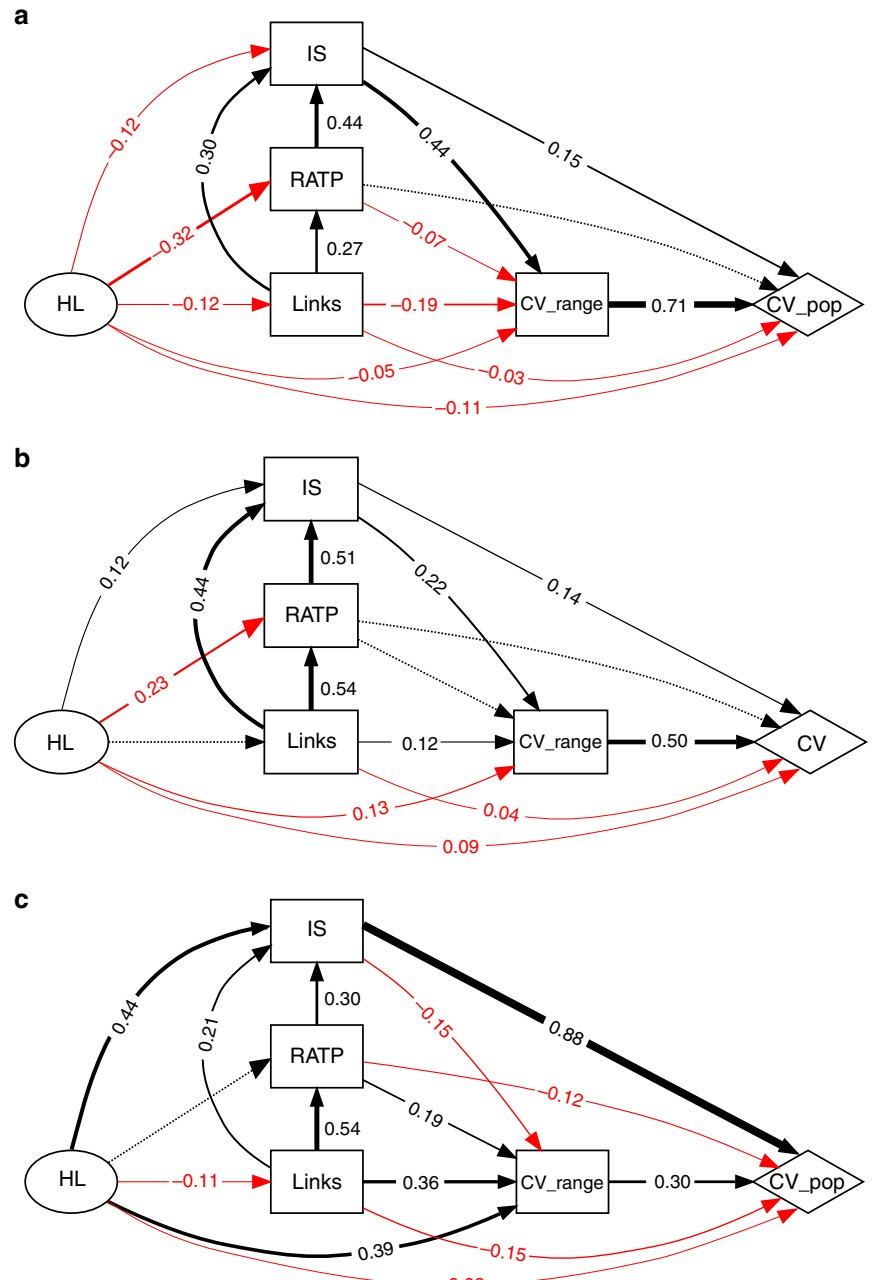

**Fig. 4** Structural equation models (SEMs). SEM diagrams show the pathways that influence stability—temporal variability of population abundances and area variability—of multitrophic communities under different types of HL: **a** random, **b** intermediate, and **c** contiguous. Given the small effect to the fraction of mutualism (Supplementary Note 2), we present aggregate results produced by grouping response data for all fractions of mutualism. This aggregation increases the effective number of replicate simulations. Each arrow indicates a link from a predictor to a response variable. Positive effects are shown in black, negative effects in red. Numbers indicate effect size, given by the magnitude of the range-scaled model coefficients. Width of solid arrows indicates effect size; dashed arrows indicate non-significant links. Multiplication of coefficients along a given path can be interpreted as a measure of the influence along that path relative to other paths in the SEM

variability increases without major changes in their structure. Meanwhile, random loss of habitat fragments the landscape while altering diversity, network structure and stability. This is consistent with previous results showing that random HL has a larger negative effect on communities than contiguous loss[13,14], given the more pronounced changes in diversity and network architecture under the former scenario.

Our results suggest that differences in community responses are due to the different pathways through which HL affects the stability of communities. In general, changes in population variability are largely driven by changes in the distribution of IS, either directly (contiguous loss) or indirectly (random and intermediate loss). Further, the relationship between area loss, IS and variability is independent of any change in network architecture as observed under random loss. However, contiguous HL decreases variability by enhancing IS, while random HL does just the opposite. These contrasting responses in variability result from the different constraints that HL type—random vs contiguous—imposes on the mobility patterns of individuals within the remaining habitat. Such differences in individual mobility

**Table 2 Direct and indirect effects sizes of predictor variables on stability (CV population)**

| Predictor | Random | | | Intermediate | | | Contiguous | | |
|---|---|---|---|---|---|---|---|---|---|
| | Direct | Indirect | Total | Direct | Indirect | Total | Direct | Indirect | Total |
| HL | −0.111 | −0.169 | −0.279 | −0.086 | −0.075 | −0.161 | −0.080 | 0.392 | 0.312 |
| Links | −0.029 | 0.046 | 0.017 | −0.041 | 0.253 | 0.212 | −0.154 | 0.554 | 0.400 |
| RATP | — | 0.152 | 0.152 | — | 0.132 | 0.132 | −0.116 | 0.305 | 0.189 |
| IS | 0.145 | 0.308 | 0.452 | 0.143 | 0.115 | 0.258 | 0.878 | −0.045 | 0.834 |
| CV range | 0.705 | — | 0.705 | 0.504 | — | 0.504 | 0.295 | — | 0.295 |

Effects sizes are given by standardized model coefficients (Fig. 4), and indirect effects are calculated by multiplying coefficients along the major pathways. Significant models were selected based on AIC$_c$ and Fisher's C of the overall path model. Then each individual path is evaluated for significance, excluding those that have a $p$-value > 0.05 (F-test). Omitted values (—) are either not significant ($p$-value > 0.05) or not retained in the model structure.
HL habitat loss, RATP relative abundance of top predators, IS mean interaction strength, CV range coefficient of temporal variation in species range area

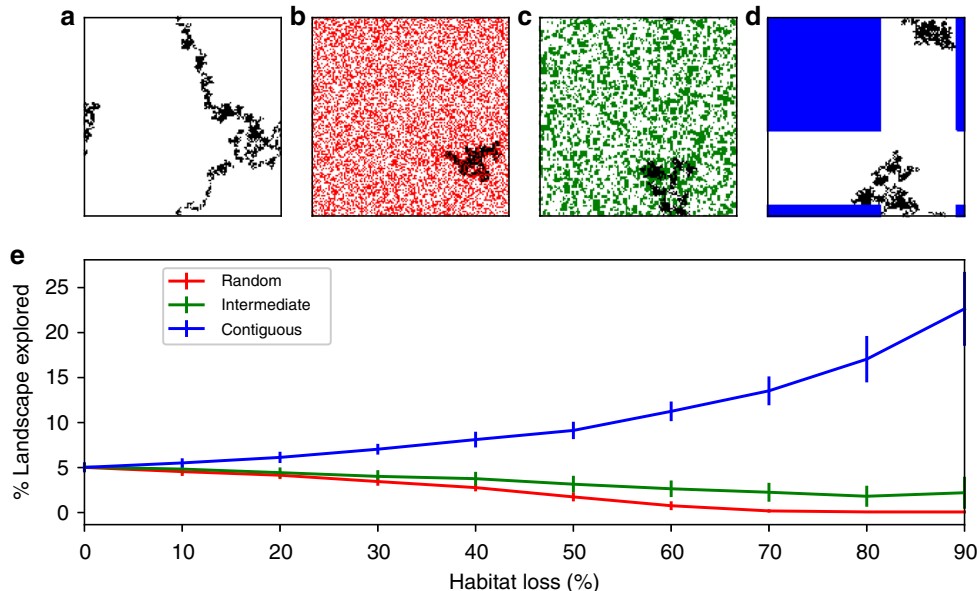

**Fig. 5** Individual movement patterns under different habitat loss types. Top row: example trajectory for a single individual over 5000 time steps in **a** pristine landscape; **b** 40% random HL; **c** 40% intermediate HL; **d** 40% contiguous HL. Pristine landscape cells shown in white, destroyed cells in red and blue for random and contiguous HL, respectively. Green cells refer to destroyed cells under the intermediate HL scenario. Bottom row **e**: Fraction of pristine landscape explored by an individual during 5000 time steps. Solid lines indicate mean over 100 repeat runs; error bars indicate ±1 standard deviation

patterns scale up to the community level and explain the differences in RATP, IS, CV range and CV population under different types of HL.

We found that temporal variability of populations increases with habitat area. The increase in temporal variability following contiguous HL is consistent with empirical observations on stability (variability)–area relationships in biological communities[26,27]. Our results give support to two mechanisms responsible for such reduction in stability under contiguous HL. Firstly, by restricting communities to smaller regions of space that have no major barriers to motion and dispersal, HL increases the encounter probabilities between interacting partners and shifts IS distributions to higher values. The negative effect of strong interactions in temporal variability reported here has been observed in mesocosm experiments[28]. Secondly, under contiguous HL the relative abundance of top predator populations remained nearly constant, and the SEMs revealed that the resulting predation pressure contributed to the loss of stability. This is consistent with the spatial-compression mechanism proposed by McCann et al.[25] in a metacommunity context, whereby pressure from mobile predators is a main driver of local variability.

Contrary to contiguous HL, random HL generates more fragmented communities in space by creating barriers to motion

which constrain individuals' mobility within the remaining habitat. This limitation on mobility at the individual level has profound consequences for community structure and dynamics. Firstly, random HL has a larger impact on diversity and network structure when compared with contiguous HL. This is illustrated by species abundances becoming more evenly distributed, predator abundance decreasing, and vulnerability and generality decreasing and increasing, respectively. Secondly, low mobility reduces encounter rates between individuals and pushes communities towards lower IS, which in turn increases the variability of their populations. Paradoxically, communities become less variable over time under random HL. However, this does not mean that random HL is beneficial for biological communities. Rather, these communities are structurally simpler and more fragmented/disconnected, and undergo extensive changes, such as the collapse of predator populations, which have been associated with higher local variability[25].

Random and contiguous loss of habitat thus relate to two different hypotheses on community responses to such disturbance: spatial compression of ecological networks and the 'empty forest' hypothesis. Actually, each type of HL represents one extreme of spatial correlation in the pattern of HL that corresponds to the two extreme cases of land use optimisation:

land-sparing and land-sharing[38]. Their joint study allows to grasp the transition from a scenario where HL and fragmentation shape the resulting communities (random HL—land-sharing) to a scenario where the effects of area loss alone (contiguous HL—land-sparing) determine the community responses to HL. Under land-sparing, biodiversity is concentrated into one or a few large habitat fragments and communities suffer from spatial-compression mechanisms that lead to higher IS and higher population variability. In contrast, under land-sharing biodiversity is distributed across the whole landscape but in a large number of smaller, fragmented patches of habitat; this has strong effects on the diversity and network structure of the fragmented communities. The strict dichotomy of land-sparing vs land-sharing has been criticised[41], and most land-use management probably lies in-between these two extremes, so that real-world communities are exposed simultaneously to the effects of HL and fragmentation. Such 'middle' or halfway responses to HL are observed for the intermediate scenario, suggesting a gradual transition of community responses to HL where elements of the spatial compression of communities as well as the 'empty forest' hypotheses are in place.

To analyse community responses to HL beyond its effects on species richness, we kept the number of species constant by having high immigration. Additional analyses with lower immigration rates showed that, if immigration is constrained, HL isolates communities by preventing colonizers to invade[42], and this eventually causes loss of species within habitat fragments. Importantly, despite species extinctions, the general trends in IS and the temporal variability of population abundances were not qualitatively affected by lower immigration rates (see Supplementary Note 4). Collectively, these results suggest that, in order to understand community responses to HL, it is fundamental to understand the type of HL as it can affect communities in diverse, sometime opposite, directions.

Two additional results emerge from our model. Firstly, the fraction of mutualism did not qualitatively affect the response of hybrid communities to HL. Although the magnitude of effect sizes differed, the stabilizing effect of mutualistic interactions revealed in recent theoretical studies[21,39] was not detected here, nor the destabilizing influence of mutualism[30]. However, we must take this result cautiously. On one hand, our study and others measure stability in different ways—we focused on population variability (as in refs. [21,26]), while others looked at the stabilizing role of mutualisms by focusing on persistence or asymptotic resilience[10,40,43,44]. Despite stability metrics can be correlated in undisturbed communities[22], these correlations can break apart under disturbances (e.g. HL) and comparisons between different stability components are increasingly difficult to establish for such disturbed communities. On the other hand, the fraction of mutualism in our study refers to the proportion in relation to herbivore links in the second trophic level rather than to the whole set of interactions in the community (e.g. random distribution of interactions across the network[39]). A 100% of mutualism does not mean that all interactions in the community are mutualistic (when all trophic interactions are replaced by mutualistic interactions the average fraction of mutualism is 24.29 (sd = 6.68%); this is representative of recent studies that consider the meta-web of species and interactions[20], although numbers may vary depending on habitat type[18]), and therefore we cannot rule out the possibility that mutualistic interactions at or between other trophic levels have a significant (de)stabilizing effect on community stability.

Secondly, we did not observe any relationship between network structure and variability. This is surprising as studies of communities with single interaction types have found that mutualistic and trophic communities respond differently to HL, especially in terms of nestedness and compartmentalization[5,6]. Yet, neither property changed consistently under any HL scenario. Similar results have been reported in recent studies of non-perturbed hybrid communities[19,21], suggesting that the lack of relationships between such network patterns and temporal variability is not derived from perturbations per se, but from the simultaneous consideration of trophic and mutualistic interactions.

Our approach has several caveats. Firstly, we mainly focused on two extreme scenarios of HL. Despite these two types of HL were chosen to represent the land-sparing vs land-sharing framework and to make results comparable with most studies on HL, an exploration of a wider range of HL scenarios would be relevant from a theoretical perspective. It would allow investigating the transition from one regime to the other and to more specifically disentangle the effects of fragmentation on biological communities. The results for the intermediate scenario, however, are consistent with a gradual transition from random to contiguous loss, as the response of diversity, network and stability metrics lies halfway between these two extreme cases. Secondly, this study follows recent claims to integrate trophic and mutualistic interactions simultaneously, but other types of interactions exist—namely competition, commensalism, amensalism—that may be affected by HL, e.g. inter-specific competition might increase along the gradient of HL. Although competition for space is implicitly considered in our model, future studies investigating hybrid communities would benefit from including additional types of interaction. Thirdly, our model depicts intensively-managed terrestrial ecosystems where destroyed habitat is unsuitable for biodiversity. However, there are other types of ecosystems (e.g. aquatic) and other management scenarios (e.g. organic farming) with different characteristics that allow certain species to inhabit fragmented habitats. In these cases, the effects of HL on biological communities may differ from the ones reported here. Our model does not consider differences between species in terms of their home ranges (HL would more strongly affect top predators as their home ranges are larger) or habitat preferences (edge vs core species[45,46]), nor it considers intra-specific variability in immigration rates, which can have a strong effect on communities[47]. Although the outcome of immigration in our model can vary among individuals (Supplementary Note 1) and varying immigration rates did not significantly alter our main results (Supplementary Note 4), future studies should address inter- and intra-specific differences in demographic parameters. Finally, because there is no experimental data to inform all parameters in the model, we selected values that produced realistic community patterns (e.g. log-normal rank-abundance distributions, exponential degree distributions; Supplementary Figs. 13, 14) and stable dynamics[21], and that reflected ecological realism (e.g. assimilation rate higher for plant than animal biomass). Sensitivity analysis shows that our results are robust to variations in the value of each model parameter (Supplementary Note 3), yet future extensions of this and other individual-based models should account for experimentally-tested values as they become increasingly available. Despite these limitations, our model is a very useful step towards a better understanding of the effects of perturbations on several levels of biodiversity, and it reveals important mechanisms of how HL affects the relationship between interaction patterns and stability of multitrophic terrestrial communities.

Our study aligns with recent efforts to assess changes in ecological communities beyond species extinctions[3,48]. By controlling the number of species, we were able to disentangle the effect that HL has on community structure from its effect on community composition. HL not only reduces the abundance of species, but may also lead to changes in several aspects of the structure and stability of communities prior to extinctions. Our

work relates with research exploring changes in local diversity following global change, currently under a heated debate[49]. We suggest that, irrespective of a positive, negative or neutral change in local diversity, the type of HL changes the structure and dynamics of ecological communities in very different, contrasting ways. Finally, our findings suggest that conservation efforts focusing on alleviating the effects of loss of natural habitats would benefit from (i) including several aspects of community structure and stability, in addition to species persistence, into assessment and management plans, and (ii) integrating the type of HL into conservation planning, as it can strongly determine the response of communities.

## Methods

**Individual-based model**. We use an individual-based, bio-energetic model developed to simulate dynamics of 'hybrid' ecological communities in a spatially-explicit context[21], and extend it to investigate the response of communities to HL. We use the term 'hybrid' not as a description of a modelling framework (e.g. models with both deterministic and stochastic components), but as a definition of communities combining trophic and mutualistic interactions. In this section, we present the key features of the model. Full model specifications can be found in Supplementary Note 1.

Several reasons justify our choice of an individual-based model (IBM) over ordinary differential equations (ODEs). Firstly, IBMs are better suited to investigate different HL scenarios more thoroughly given their spatially-explicit nature. The IBM used here accurately describes (in terms of rank-abundance and degree distributions[21]) the structure and dynamics of non-disturbed communities. Secondly, IBMs constitute more intuitive modelling frameworks when one is interested in scaling up processes happening at the individual level (e.g. individual movement across the landscape) to patterns at the community level (e.g. network properties, community stability). A third reason to choose IBMs is that we want to include variability at the individual level in our model, so that every individual can potentially be at different physiological or bio-energetic states at any given time step or location. This will turn affect its demographic activity (e.g. reproduction, death, etc). In addition, technically, it is virtually impossible to model the dynamics of thousands of individuals with ODEs, as this would require a very large number of equations, and analytical solutions would be elusive. IBMs are better suitable to include such elements, and therefore we prefer them over ODEs to accomplish the specific goals of this study.

The model does not explicitly consider any spatial scale. This is because our goal is not to represent faithfully a particular terrestrial community or specific ecosystem, but to investigate the response of a standard/ideal community to the loss of its habitat. Defining spatial units more specifically is highly dependent on the system studied, and would make model parameterization much more complex. For similar reasons, the model does not consider a specific temporal scale. The model allows consumers to always take resources, and this is a realistic assumption given that all individuals spend energy in each time step according to bio-energetics. This assumption of the model implies that the length of a time step can be seen as the time at which an individual spends a sufficient amount of energy as to feel the need to find more resources.

**Community dynamics under habitat loss**. The model is fed with an interaction network which defines the potential interactions between pairs of species in a simulated ecosystem. This network is generated in two stages. Initially, the niche model[50] is used to produce a food-web displaying structural features similar to those observed in nature. Subsequently, a given fraction of the trophic links between primary producers and second trophic level species are turned into mutualistic links in order to obtain an ecological network with trophic and mutualistic interaction types. This is equivalent to, for example, replacing some herbivorous interactions with pollination interactions. Therefore, our networks reproduce some of the most studied mutualistic interactions, such as seed-dispersal and plant–pollinator interactions, embedded in larger ecological networks. The fraction of links thus replaced is referred to as the fraction of mutualism. Varying the fraction of mutualism allows us to determine the role of mutualistic interactions in mediating community responses to HL. Although we could have included mutualistic interactions in a configuration that is more typical of mutualistic networks (e.g. binary nestedness pattern), we prefer to be consistent when generating the full network of species interactions. Further, the niche model describes trophic niche occupancy between consumers and resources along a resource axis and successfully generates network structures that approximate well the central tendencies and the variability of a number of food-web properties. Because it arranges consumers and resources along a resource axis, the niche model can be applied to other types of consumer–resource interactions (aside from antagonistic predator–prey interactions)[50].

Once the interaction network is established, community dynamics driven by bio-energetics, spatial constraints, and species interactions are then simulated through time using an individual-based model[21]. Our model departs from others in the following ways: (i) individuals within species have different extinction rates not dependent on stochastic processes (i.e. they are independent from any pre-defined probability), which eliminates the need to define fixed extinction probabilities for all species in the community, (ii) more complex demographic processes such as reproductive ability and immigration are taken into account, (iii) bio-energetic constraints, such as energy transfer efficiency and energy loss at the individual level, drive population dynamics of species in the community. Stochasticity is modelled based on pre-defined probabilities; these probabilities at every time step are obtained using a pseudorandom number generator provided by Python. The model has been demonstrated to generate in silico communities that quantitatively resemble empirical ones in terms of rank-abundance and network degree distributions (Supplementary Figs. 13, 14)[21]. To disentangle the effects of HL from those associated with changes in species richness, we control for the number of species by including a high, yet realistic, rate of immigration from a regional species pool. This mechanism is similar to the 'rescue effects' common to metacommunity modelling[42]. Therefore, the model is representative of communities open to immigration, rather than closed communities, such as remote island systems. The dependence of model outcomes on immigration rates is tested with sensitivity analysis (Supplementary Note 4).

The landscape consists of a homogeneous squared two-dimensional lattice ($200 \times 200$ cells) on which individuals move around and interact subject to bio-energetic constraints. To avoid edge effects, the lattice has periodic boundary conditions such that the topology of the landscape is toroidal, i.e. the sides of the lattice are connected, thus creating a continuous space. Each lattice cell has a space for an inhabitant and a visitor, such that a cell may contain at most two species. Basal species (plants) may only occupy empty, available cells, whilst all other species may occupy both empty cells and cells already occupied by a plant. For simplicity, available, non-destroyed cells do not differ in habitat quality. Initial conditions are defined randomly (and independently for each replicated simulation) via the following procedure: for each cell in the landscape an individual belonging to a randomly selected basal species is placed in the inhabitant space, so that all cells with a plant individual. Then individuals from randomly selected non-basal species are placed in the visitor space of randomly selected cells, until the desired fraction of the landscape (given by parameter OCCUPIED CELLS, see Supplementary Note 1) is filled with animal individuals. Simulations are then run following the local demographic and interaction rules described in section 1.1, Supplementary Note 1.

During each simulation, undisturbed communities with stable dynamics are first generated. Once a stable community is obtained, the landscape is perturbed by destroying a specified fraction of the lattice cells (the intensity of HL), in successive steps (% of HL), according to one of two HL scenarios—random or contiguous—which represent the two extreme cases of optimising land use—land-sparing vs land-sharing[38]. This also allows our model results to be comparable with other results, as previous studies have modelled random[30–32], contiguous HL[33–35], or both[13,36,37]. Disturbed communities at each fraction of HL are let to evolve through transient phase. Subsequently, the model computes the metrics for that given fraction of HL. Under the random scenario, lattice cells to be destroyed are chosen uniformly at random from the set of remaining cells. Under the contiguous scenario, an initial seed cell is chosen at random, from which the loss proceeds radially outwards to produce a contiguous region of destroyed habitat. We further consider a third intermediate scenario that corresponds to a degree of spatial correlation of 0.5 (halfway between random and contiguous loss). Once a lattice cell is destroyed, it can no longer harbour any individuals or be colonized. Thus, this represents intensively-managed landscapes where biodiversity is prevented from perturbed fragments. Model simulations consist of 5000 time steps or updates of each cell in the landscape according to the rules of engagement between individuals (Supplementary Note 1). For each HL scenario, we ran simulations at 11 different fractions of mutualism (range: 0.0–1.0), and 10 different intensities of HL (range: 0–90%), with 25 replicate simulations for each pairwise combination of fraction of mutualism and HL, yielding a total of 2750 simulations. Each simulation is initiated with a unique random placement of individuals for each species from the community, and a uniquely generated interaction network structure.

We expected that the two extreme HL types to have contrasting effects on the spatial configuration of the remaining habitat, and this may in turn influence individual mobility across the landscape. Individual mobility therefore reflects the realized effects of fragmentation, with individuals exploring a smaller fraction of the landscape when fragmentation is high and vice versa. To investigate the effects of HL type on the average movement pattern of individuals, we run 100 additional simulations for 5000 time steps with a single individual belonging to a consumer species, at each level of HL and for all types of HL. We quantify individual mobility as the average fraction of the pristine landscape cells visited during the simulation time.

**Community responses to habitat loss**. To characterize communities and study their responses to HL we quantify three categories of community metrics describing diversity, network structure and stability properties. We list them below and give full definitions in Table 1.

*Diversity*. We measure total abundance, and different features of the species abundance distributions. We use the Shannon diversity and equitability indices to

measure how evenly abundance is distributed between species, and quantify the relative abundance of top predators (RATP) as an indicator of the density of top predator populations.

*Network structure.* The realized network of interactions is evaluated at the end of each simulation by tracking the interaction events between pairs of individuals during the final 200 time steps of each simulation. We do this to avoid misleading results due to transient dynamics (the last 200 time steps are considered to be 'stable'). We measure several network properties which are commonly used in ecological networks studies: number of links (i.e. realized interactions between pairs of species), compartmentalization[51] and nestedness[52]. We also account for changes in interaction frequencies (i.e. number of recorded encounters between individuals of two potentially-interacting species) which are not reflected on topological modifications by evaluating properties such as the quantitative generality ($G_q$) and vulnerability ($V_q$)[53]. In addition, we compute summary statistics of interaction distributions by calculating the mean inter-specific interaction strength between species (IS)[54], since there is strong evidence that interaction strengths are related to population variability[28].

*Stability.* We quantify two stability metrics during the final 200 time steps of each simulation (to avoid transient phase): (i) temporal variability, a metric that has been extensively used in both the theoretical and empirical literature[23,55], and measured as the average (across species) of the coefficient of variation in population abundances through time (CV population); and (ii) area variability, proposed by Lurgi et al.[21], measured as the average of the coefficient of variation in species range area through time (CV range). On one hand, we opt for CV population because there is evidence that interaction strengths may be altered by HL which we hypothesise to be associated with changes in variability. Although asynchrony in the abundance or biomass fluctuations across trophic groups can contribute to stabilize ecosystem functioning[56], it may mask inherent variability when adding population time series to obtain the community biomass dynamics. Therefore, we consider the use of species level variability is more appropriate than community level variability in this multitrophic context (see ref.[55] for clarification of these two concepts). On the other hand, HL affects individual mobility, which may affect the spatial range of species and the strengths of species interactions; CV range captures the average variability of species' range areas in the landscape. It is worth noting that the response of CV population and CV range to HL may differ; for instance, a change in the population density of species may alter its spatial range without an associated change in its abundance.

**Statistical analyses and robustness.** To test how communities respond to HL, we fit linear models to the relationship between HL and each of the response metrics described above. These models are fitted using the package *statsmodels* in Python[57]. Variables that respond non-linearly to HL were transformed before linear fitting (these transformations are specified below where relevant). From each significant linear model, we obtain estimates of the direction and magnitude of the response across the HL gradient. We refer to the magnitude of the response as the 'effect' size. In order to further assess differences of community responses across HL types, redundancy analysis is performed on the standardised (scaled to zero mean and unit variance) values of all response variables except for number of species, compartmentalisation and nestedness (due to their weak or null response to HL). Thus, redundancy analysis is performed over a number of individuals, Shannon diversity and equitability, number of links, generality, vulnerability, RATP, CV population, CV range and mean interaction strength (IS) using HL type as the constraining variable. Only community responses to HL fractions equal or larger than 50% (i.e., between 50 and 90% habitat lost both inclusive) are included in the analysis to capture strong responses across the measured variables and thus highlight the differences across HL types. HL type (random, intermediate, and contiguous) is used as the constraining factor in the redundancy analysis. This analysis is performed using the *rda* function from *vegan*'s R library.

Due to the large array of variables measured, and the scope for collinearity amongst them, we use piecewise structural equation models[58] (SEMs) to provide a more mechanistic understanding of the trade-offs, feedbacks, and other interactions among diversity, network structure and stability. The focus of the SEM analysis is on disentangling the potential mechanisms driving changes in stability under HL. Since the results from the SEMs are not qualitatively affected by the fraction of mutualism (Supplementary Note 2), we combine the simulation results for all fractions of mutualism into a single dataset, effectively increasing the number of replicate simulations. The full details of the methodology and underlying hypotheses for the SEM analysis are given in the Supplementary Note 2.

The parameter values used come from Lurgi et al.[21], where this model is introduced and shows that simulated communities display patterns similar to those observed in real communities. Some of these quantitative patterns include log-normal rank-abundance distributions and exponential degree distributions (Supplementary Figs. 13, 14). Certain parameter values are also based on ecological realism, e.g. assimilation rate is higher for plant biomass than animal biomass (*HERB EFFICIENCY* > *EFFICIENCY TRANS*). However, in order to explore model robustness to the variability of parameter values, we perform a sensitivity analysis using latin-hypercube sampling[59]. In general, our results are robust to variations of

±20% in the value of each model parameter. The details of model parameters and the sensitivity analysis are provided in Supplementary Note 1 and 3.

**Reporting summary.** Further information on research design is available in the Nature Research Reporting Summary linked to this article.

## Data availability
The data that support the findings of this study are available from the corresponding author upon reasonable request.

## Code availability
The simulation code that supports the findings of this study is available at https://github.com/cm1788/Stability-of-multitrophic-communities-under-habitat-loss (https://doi.org/10.5281/zenodo.2634231).

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

## Acknowledgements

We thank the computational facilities of the Advanced Computing Research Centre at University of Bristol (www.bristol.ac.uk/acrc). C.M. was funded by EPSRC through the Bristol Centre for Complexity Sciences. DM was funded by the EU and INRA in the framework of the Marie-Curie FP7 COFUND People Program, through the award of an AgreenSkills/AgreenSkills+fellowship. AMCS was supported by the French ANR through LabEx COTE (ANR-10-LABX-45). D.M., M.L. and J.M.M. are supported by the French ANR through LabEx TULIP (ANR-10-LABX-41; ANR-11-IDEX-002-02) by a Region Midi-Pyrénées Project (CNRS 121090), and by and the FRAGCLIM Consolidator Grant, funded by the European Research Council under the European Union's Horizon 2020 research and innovation programme (grant agreement number 726176).

## Author contributions

D.M., M.L. and J.M.M. conceived the original idea and designed the research. D.M., J.M. M. and M.L. designed the model, with help from C.M. C.M. performed the analysis. C.M. and D.M. wrote the first draft of the manuscript, all authors contributed substantially to revisions.

## Additional information

**Competing interests:** The authors declare no competing interests.

