## [Peer Review File · Nature Communications]

Reviewer #1 (Remarks to the Author):

Review of "Routes to instability under habitat loss: an investigation of multitrophic communities" by McWilliams et al.

I reviewed a previous version of this manuscript for a different journal, and provided a number of comments and suggestions to improve the manuscript (included below). I am disappointed to find that none of my suggestions have been incorporated into the current version. In fact, the only difference I can find in the current version that the Results section has been moved before the Methods section; all text appears again verbatim. (I was not able to see if changes had been made before accepting to review the current version.)

I spent a good amount of time reading and thinking about the manuscript in the hopes of improving both the science and its presentation, and encourage the authors to consider at least some of my suggestions.

Although previously I was supportive of publication, the lack of desire of the authors to make even the simplest of changes to improve the manuscript leaves me decidedly neutral about this work. (For example, in my previous review I noted that the authors had mistakenly included two consecutive "of"s in Table 1 in the row about "nestedness." I see that this error is still present.) I have said my piece in the previous review and offer those comments to help with the decision-making process at Nature Communications.

[Previous review of the present manuscript]

Review of "Routes to instability under habitat loss: an investigation of multitrophic communities" by McWilliams et al.

The authors propose an individual based model of community dynamics to investigate how habitat destruction affects the stability of ecological communities comprising multiple interaction types. They consider two types of spatially-explicit habitat destruction: successive loss of random and contiguous habitat patches; and two types of interaction: trophic and mutualistic interactions. They assess stability in two ways: temporal variation in species population abundances and temporal variation in species range area. The authors use their modelling approach to show that communities are more affected by random habitat loss, and that corresponding changes in stability are insensitive to the fraction of mutualistic interactions in the community.

The manuscript is clear and well-written, and addresses an important question in community ecology: does modelling multiple rather than single interaction types change how we expect communities to respond to habitat loss, and will responses vary depending on the form of habitat loss? The authors propose an interesting modelling approach to begin exploring this and related questions. I have three major comments about the work: 1. Can we definitively rule out the counterclaim that mutualism does have a qualitative effect on community stability? 2. The modelling approach should be expanded to allow for a more general and flexible investigation of community responses to habitat loss. 3. Some language should be clarified to avoid confusion. If these comments could be adequately addressed then I would be supportive of publication in [JOURNAL NAME].

1. Can we definitively rule out the counterclaim that mutualism does have a qualitative effect on community stability?

The authors build ecological networks with multiple interaction types in two steps (P8L31): first, they build a food web using the niche model, then they replace a fraction of the trophic interactions between the two lowest trophic levels by mutualistic interactions. The authors then vary the fraction of *replaced* interactions between 0% and 100%.

Before I accept the authors' claim (P3L18) that results on stability are insensitive to the fraction of mutualistic interactions in the community, I have two questions:

At 100% replaced interactions, what is the *total* fraction of mutualistic interactions to trophic interactions?

Presumably their communities have more than two trophic levels, so even with 100% replaced interactions the overall fraction of mutualism in the community could still be very small. In which case, it would be unsurprising that a small number of mutualistic interactions had little effect on community stability.

What happens if interaction types are distributed randomly throughout the network, or model networks are based on distributions of interactions based on empirical communities with multiple interaction types?

Given that this work is largely a theoretical contribution, I would like to see a more comprehensive study of possible cases (even if some of the results end up as supplementary information).

Finally, the authors consider trophic (+-) and mutualistic (++) interaction types. For completeness, it would be interesting to also consider competition (--).

2. The modelling approach should be expanded to allow for a more general and flexible investigation of community responses to habitat loss

The authors consider two versions of habitat loss: random and contiguous. It would be useful to present a more general model that can describe a wider range of possible habitat loss scenarios. For example, the authors could have a single parameter (h) that controls how habitat patches are selected for destruction at each time step. Values for h would range from perfect anti-correlation of selected patches ($h = -1$), to slight anti-correlation ($h = -0.5$), to random ($h = 0$, one of the two currently studied cases), to slight correlation ($h = 0.5$), to perfect correlation ($h = 1$, the current, contiguous, case).

Analyses could then explore how community responses to habitat loss vary along a continuum. This would represent a more helpful theoretical baseline than the simple dichotomy of random and contiguous habitat loss. It would also permit analysis of how patch-size distributions are related to community responses.

3. Some language should be clarified to avoid confusion

The authors use the term "stability" throughout the manuscript, but the term is used to refer to very different concepts. This is unavoidable because researchers have used many different technical approaches to study "stability" (as mentioned, P6L26). To minimise confusion, I suggest the authors define how *they* measure "stability" earlier in the manuscript. This could be done in the Abstract (P3L16), and in the paragraph introducing their work (P7L17). The authors should also be careful not to make comparisons between results for different definitions of stability (P20L21, P21L22) without explicitly mentioning which definitions are being used.

Given the prominence of "stability" in the manuscript, the authors should consider, perhaps in the Discussion, how results for *their* definition of stability fits in with previous or future work that addresses a similar question but with a different measure of community "stability".

The authors do not demonstrate any kind of causality in the study, so I suggest avoiding the use of this term. Even within the context of their models (for which establishing causality should be more straightforward than in general) they do not formally establish a causal chain (by comparing the relative importance of directly modelled processes, for example). Although I value structural

equation models, personally, I do not consider their outputs as sufficient evidence to imply causality (i.e., how can we be sure whether it is a loss of interactions that causes a change in relative abundance, or a change in relative abundance that causes a loss of interactions). Some points where statements about causality should be reconsidered include: P3L25, P3L32, P17L45, P18L37, Figure 3 caption.

The authors should clearly and unambiguously define "interaction frequencies" and "interaction strengths" (P11L35). This would serve to avoid any confusion between concepts including: i. the number of recorded contacts between individuals of two given species in a model or empirical data, ii. interaction parameters in a bioenergetic model, and iii. summary statistics of interaction distributions (e.g., P11L44 and P11L47).

Minor comments

P3L28. The "two types of habitat loss" are not introduced in the abstract.

P7L8. This sentence needs clarifying: are the authors trying to say it is not clear how habitat loss affects the *relationship* between interaction patterns and stability?

P7L54. The authors could also cite examples of mutualistic interactions *destablising* ecological communities, e.g., Staniczenko et al. (2013) The ghost of nestedness in ecological networks. Nature Communications, 4, 1931

P8L45. "...replacing some herbivorous interactions..."

P9L42. There needs to be a paragraph here explaining how species are initially distributed across the landscape (and how many individuals per species and the "shape" of the landscape and how boundary effects are handled); some of the text from P10L32 could be moved earlier.

P10L35. How do mobility values vary between individuals and species? Do individuals perform a random walk?

P13L49. Although I like the headings in the Results, should they match the three categories introduced on P11?

P14L3. Are the authors meaning to suggest that random habitat loss is a good thing? Or do they mean that communities were less unstable under random habitat loss compared to contiguous habitat loss (although this contradicts the previous sentence)?

P14L26. Are there identifiable patterns with contiguous habitat loss?

P14L31. Clarify how links are lost: is it due to no co-occurrence between species in a grid cell, or a *shift* away from an interaction even though the species do co-occur?

P16L3. Mention that results in this paragraph are from linear models.

P16L40. Explain numbers like "0.71".

P16L51. How exactly does this statement relate to the results just presented? Explain *how* the distributions of interaction strengths are changing, and how this affects stability in the two habitat loss scenarios.

P17L17. Clarify this sentence.

P17L51. What is "the link between habitat area, interactions [sic] strengths and stability" exactly?

P18L10. It would be helpful if the definition of stability in terms of variability in population abundances was restated.

Table 1. In "nestedness", there are two "of"s.

Table 2. Are all values significant, and, if so, at what level?

Figure 1. What do the numbers on the scale mean?

Figure 2. What about CV_range?

Reviewer #2 (Remarks to the Author):

McWilliams et al present an interesting simulation on the effect of habitat loss on community stability. The work presented here is based on extensive previous knowledge and a well-developed individual based model yet it presents a novel and timely extension by including the effect of habitat loss and including two types of interactions. The results are in general well discussed and presented, although at some points the text is difficult to follow. However, although I believe the work is a great contribution I have several concerns regarding the applicability and usefulness of these approaches to real-world settings and also am afraid that the contents of these paper might be better suited for a more specialised journal.

The authors suggest in their model that once a cell is destroyed during the habitat loss phase then no species can move there and no offspring can be placed there. I understand this is a simplified model of reality but wonder if it wouldn't be more realistic if some species were allowed to use these destroyed cells, given that many species are able to live in the matrix of newly fragmented habitats. I wonder whether not allowing for this has an important effect for your results. Also, there is no consideration of edge effects in the model and how they could affect the community dynamics. In particular, given that one of the interactions considered is the mutualistic interaction of seed dispersal, edge effects are particularly important as in many cases for example more fruits are produced in edge habitats.

The authors present the two scenarios of habitat loss as the two extremes that we might expect in nature, yet they are not equally common, rather the contiguous scenarios tends to be the norm while the random one would be something rare. Even in fragmented areas where habitat loss is not contiguous, it is also not random. The authors could comment their results in light of this difference between the probabilities of scenarios taking place. Also, the authors could place these two scenarios within the available literature, whether it's the fragmentation or the land sparing-land sharing or whichever they choose. At the moment the reason why these two contrasting scenarios were chosen and what their implications are is diluted throughout the text.

The text is difficult to follow sometimes as the acronyms used and many of the concepts are not described before they are used the 1st time. For example, it is unclear what MAI means but it seems it might be related to FM, fraction of mutualists? In line 126 you start to use HL for habitat loss but you have never introduced it before and the reader has to assume that is the case. Then you indistinctly use "habitat loss" or "HL" and "IS" or "interaction strength", stick to one of the two and be consistent.

The abstract by itself is very difficult to follow. For example, it is not clear when your first read it what the authors mean by "hybrid communities". Neither is it clear what they mean here by "two types of habitat loss". I understand that given word constraints this is not always an easy task but the abstract alone should be able to tell the reader what the paper is about and what the general conclusions are.

In the Introduction L63-74, the authors describe habitat loss and its effects, yet they tend to mix concepts. First, they are comparing random or spatially-correlated habitat loss and then they

introduce habitat fragmentation, which entails habitat loss but also many other effects (edge effects, loss of connectivity), which this paper in particular does not take into account.

Across the whole paper and in the Abstract and Intro sections the authors claim that their study addresses how communities “respond to habitat loss prior to species extinctions”. However, the number of species does not remain constant throughout the simulation, but rather some species go locally extinct (as shown in negative effects of habitat loss on number of species on Fig. 1), so I am not sure this is correct. Maybe I have not understood properly, as the authors suggest this is countered by high immigration rates. But at present I believe this needs clarification.

The authors introduce their hypothesis at the end of the Intro section yet some of their expectations are not very elaborated. For example, why do the authors expect that the fraction of mutualism will affect stability? What kind of negative effects are expected from Hypothesis 2?

Sometimes the text is hard to follow. For example:

L137-140: “The contrasted responses of network...suggesting an important difference...” A difference compared to what?

L140: “For some of these properties...” Which properties?

L169: here you are referring to the results of a previous study, not the one presented here. Please specify this.

L171: Similar how?

L186: Explain what SEMs are, it's the 1st time you mention them.

L191: which is the latter case? This sentence is unclear

You use habitat destruction and habitat loss interchangeably, please stick to one terminology consistently.

L208-243 The first part of the discussion basically repeats the results already enumerated in the results section with no implications or explanations to the observed patterns.

L254: the authors suggest that because the abundance of top predators remains constant under the contiguous habitat loss scenario, interaction strengths increase and this results in greater variability. However, they have not included home ranges for any of these top predators, and the decrease in habitat available might mean some might go extinct because their home range exceeds this habitat. Also, as habitat available decreases competition between species might increase, yet this has not been taken into account either. A list of caveats would be a good way to proceed.

L253: “...increased abundance variability in forest fragments” you are discussing the findings of the contiguous habitat loss scenario in the light of the fragmentation literature, yet fragmentation would be more in the lines of the random scenario in your case.

L260-261: please re-write this sentence, it makes no sense.

L262-264: that's because you assume the matrix is absolutely unsuitable even for movement between habitat areas, which is not realistic in many cases.

It seems like the Methods section originally followed the Introduction and was then shifted because reading the Methods at the end explains many things that are not clear as the paper is presented right now. Consider changing this.

Reviewer #3 (Remarks to the Author):

Manuscript #: NCOMMS-18-15210-T

Title: Routes to instability under habitat loss: an investigation of multitrophic communities

General comments

In this manuscript, McWilliams and coauthors present an individual-based model to study the effect of habitat loss on multitrophic ecological communities. Main novelties are the inclusion of mutualistic interactions and the evaluation of changes induced by habitat loss prior species extinction occurs. The analysis of simulation outcomes highlighted that two extreme types of habitat loss dynamics have fundamental differences on ecological communities. When habitat loss happens according to the contiguous scheme the food webs get more unstable, and this response is strongly driven by increase in the average interaction strength. If the habitat loss follows the random scheme the diversity and the structure of the community are highly impacted due to landscape fragmentation. Contrarily to expectations, the proportion of mutualistic and antagonistic interactions has almost no consequences on the patterns identified by the study. In what follows there are a series of general, more substantial concerns that should be addressed by the authors. General comments refer to the individual-based model (IBM), the structure of the introduction, the meaning of stability, and the lack of a paragraph to properly discuss the limits of the simulation approach. Other minor comments offer both methodological suggestions and editing hints.

The manuscript is well written and I appreciated the effort done by the authors to condensate in simple messages the findings of their analysis. It is well known that the appeal of IBM in ecology has often been impaired by difficulties in obtaining results of general value (i.e. IBM are often case-specific but a few exceptions exist in the literature; e.g. see Giacomini et al. 2009) and also the need of highly parameterized equations often hindered their success or caused skepticism (i.e. modelers risk to address complexity with other complexity, failing to capture key mechanisms governing systems' dynamics; e.g. see Black and McKane 2012). I believe these issues have been partially addressed by the present manuscript. The use of SEM to interpret under a cause-effect perspective the results of simulations goes in the direction of simplifying the complexity and illustrates possible general mechanisms behind the recorded responses. However, there are some concerns related to the structure of the model (type of processes simulated, equations used and parameters chosen)

and its generality (i.e. it is suitable for terrestrial systems but not aquatic ones) that should be addressed prior to consider the manuscript for publication.

More efforts should be dedicated to explain the value of such approach in the context of terrestrial and aquatic ecology. My feeling is that while the application of this IBM can be appropriate for the case of terrestrial ecology, some concerns exist in the case of marine systems. For example, how to explain the mutualism in marine systems in an analogous way as it is presented in the manuscript for terrestrial plants? There are for sure mutualistic relations in marine systems, but I can hardly envision their behavior in the sense coded by the MUT_EFFICIENCY function. Although an example of pollination in the sea has been recently described (see Van Tussenbroek et al. 2016), this is far from being representative for the marine systems and cannot be used as justification to extend the validity of the IBM to marine food webs. Another reason to define the current framework as inappropriate to model aquatic systems is represented by herbivory (HERB_FRACTION function). In the model the non-mutualistic herbivore consumes a fraction of plant energy and both individuals continue living; this is clearly not the case in most of aquatic interactions involving the planktonic community. For example, in the copepod-phytoplankton interaction the feeding relationship ends with the removal of the phytoplankton cell. What I mean is that this work seems to apply better to terrestrial systems than to aquatic ones, especially given the structure and principles governing some functions. I suggest changing the emphasis of the manuscript and presenting it as an application that can (eventually) be generalized in the context of terrestrial food webs. Other specific aspects that are not convincing about the IBM (or questions that should be addressed) are provided in the list of below:

--- Can the authors briefly explain the benefit of using IBM for investigating changes in the properties of multitrophic communities in presence of various habitat loss scenarios? Why did they choose to model the consequences of various types of habitat fragmentation on food web stability using IBM and not ODEs? Are they dealing with small population sizes? Why do authors expect stochasticity to play an important role in the processes under investigation?

--- The authors kept the number of species constant by having high immigration rates. To quantify the bias imposed by such (often unrealistic) condition, I suggest performing other simulations for comparison with current results. I am not asking the authors to completely change the structure of their manuscript, but recommend including this additional analysis as new Appendix file. Results could be used to comment the possible consequences of such change in the IBM (e.g. in a new paragraph that discusses limits of current IBM, and maybe to integrate the sentences at L283-289).

--- Is it realistic to assume that all landscape cells have the same quality? Are there other individual-based models that modulate migration in a landscape depending on the quality of various patches?

--- Did all taxa display the same migration rates (L350-353)? Or were migration rates randomly assigned even within each taxon? It would be useful to explicitly read whether intra-specific variability was considered as this might have profound consequences on the outcomes of the model (e.g. see Bolnick et al. 2011).

--- L323 = I would be careful in using the term "hybrid" here. In the framework of modelling literature "hybrid" refers to models that include both deterministic and stochastic components while here the term deals with mixed types of ecological interactions. The term should be either removed/replaced or it should be clearly stated this does not describe a type of model construction/methodology.

--- L343 = I would have expected the extinction rates of various species are stochastic (i.e. not defined with a static parameters but stochastically regulated by the execution of simulation). Am I wrong? Or maybe the authors simply refer to stochastic as dependent on a pre-defined probability? Clarification is needed.

--- L345 = Are demographic processes modelled as stochastic? More generally, how is stochasticity generated (i.e. using deterministic pseudorandom number generation – e.g. analogous to what can be obtained using the `set.seed` command in R – with Monte Carlo method or a variety of it – e.g. Gillespie's Stochastic Simulation Algorithm)? The way stochasticity is generated should be explicitly declared.

--- L366 = What does one step correspond to (i.e. one hour, one week, one month or one year)? Is it realistic assuming that all taxa have same immigration probability (see Table 1 in Appendix S1) and move with the same velocity (i.e. one cell per step of simulation, if the recipient cell is not occupied by other consumers or excluded from the simulations due to habitat loss)?

--- L397 = Which is the rationale behind characterizing network structure during the final 200 steps? Why not the last 100 or 1000 steps? Were simulations exposed to transient phase during the progressive habitat loss and the authors wanted to get rid of this effect? Were the last 200 steps considered to be stable and with no habitat loss occurring? More details need to be provided in the manuscript.

--- Appendix S1 = At page 2 it is stated that “cell update consists of the following ordered processes...” Does it mean state variables are updated in an asynchronous way (i.e. the new values are not stored until all individuals have executed the process, but updated in a sequential way within each time step – e.g. this can have an effect on when trophic interactions occur; see also Appendix S1 at page 4)?

--- Is there an effect of satiety for feeding interactions? I could not find this aspect mentioned in the Appendix S1. Please, add one sentence on satiety to say whether it is considered or not.

--- The way the parameter space was validated (i.e. “...a set of parameter values were selected that produced realistic community patterns and stable dynamics.”) does not seem properly documented. Do the authors also refer to the sensitivity analysis of Appendix S2? I expect some more emphasis on this choice in the main body of the manuscript, especially considering that default values of model parameters (Table 1, which should be Table S1) were not obtained from real empirical data. Also, what does exactly mean that “realistic community patterns and stable dynamics” were obtained? This is too vague and a more quantitative justification is needed.

--- Appendix S1, at page 4 = Why the choice of offspring placed in a range of 3 cells of distance? Which is the distance in meters? Are there literature references to support such choice? How is the choice of when to release the seed taken by the herbivore in case of mutualistic interactions (i.e. in terms of number of steps)? Are these choices mediated by probabilities? Were these probabilities estimated using empirical data? If yes, literature references should be indicated. More in general, the correspondence of cells with spatial scale (one cell = 1 square meter?) needs to be supplied.

--- Appendix S1, last page = My understanding is that at each simulation step of the random habitat loss one cell is destroyed. Is this correct? Does simulation runs last well beyond the moment the last cell is removed? If yes, for how long? Why (is this related to the need of attaining a stable condition after the transient phase)? Does the same condition (i.e. one cell removed per time step) apply to

the case of contiguous habitat loss? If not, can these differences in cell removal scenarios between the two types of habitat loss affect the results?

--- Last line of Appendix S1 = Sometimes the information provided is technically correct but does not help non-experts to understand the reasons of the choices taken (e.g. when talking about toroidal boundaries that, I guess, were considered to avoid edge effect during simulations). I think that more explicit and intuitive explanations should be given, especially if the goal is being of appeal for a wide audience of ecologists.

The second aspect that should be improved in the manuscript relates to the structure of the introduction. There are poorly defined concepts and the adoption of some definitions strikes with mainstream classifications in ecology. Sometimes, the manuscript is biased towards the modelling perspective and risks to be detached from terms and definitions that are common in experimental and theoretical ecology. The contribution of the authors is valuable but they should not degrade ecology to modelling. Also in discussions and conclusions I would like to read more considerations about the ecological value of their findings, rather than general statements that are poorly coordinated with the rest of the manuscript (e.g. at L311-312 they first presented possible value of the IBM to explore the role of global changes in shape local diversity – this seems to be poorly coordinated with the rest of the manuscript and could be removed). I provide below a list of points that should be taken into account to improve the introduction:

--- L67-69 = I suggest the sentence should be rewritten. I do not understand how “extinction thresholds are higher when habitat loss is spatially-correlated” (compared to random loss) “because spatially-correlated destruction leaves larger areas of pristine habitat intact”. If larger areas of pristine habitat are intact I would expect less extinction. Maybe the confusion is related to the use of the terms “extinction thresholds”. Anyway, clear statement of the causative mechanism is required in the text.

--- L78-80 = I was a bit puzzled by the definition of food web proposed here. Food webs depict trophic interactions (no matters whether they are plant-herbivore or predator-prey); I would replace the terms “food webs” with “ecological networks” or “network studies of ecological communities” to better accommodate also the plant-pollinator interactions.

--- L88-92 = The concept of stability is central for this manuscript. While I prefer a mathematical definition of stability, which cannot be applied to this study (as it requires the presence of ODEs; see Neutel et al. 2002), I understand the need of the authors to rely on different ways to quantify stability. Nevertheless, they should be more explicit since the beginning about the meaning of “stability”. First, which are the other aspects of stability that can be affected by habitat loss (L89)? Mention these alternative aspects before citations. Second, the various definitions of stability are correlated. For example, more extinction events often result in higher levels of spatial heterogeneity and change the structure and functioning of the food web. This is the case of meso-predator release due to top-predator extinction from local patches, which is triggered by habitat fragmentation in coastal southern California (Crooks and Soulé 1999). Also, changes in food web architecture (e.g. reduction of interactions) can sharpen the risks of secondary extinctions following primary extinction events (Allesina and Bodini 2004). These aspects should be made explicit in the text.

--- L109-113 = Which is the way used to quantify stability (e.g. stability in the food web structure, in the number of species or related spatial and temporal heterogeneity)?

--- L114-118 = The meaning of “types of habitat loss” (intended as dynamics following either random or continuous patterns) gets clear only at the end of the introduction. I am convinced the reader would benefit of an earlier clarification of this concept, which is central for the manuscript.

Clear definition of interaction strength should be stated as there are many strategies for quantifying the strength of interactions (see Berlow et al. 2004). The definition of strength is essential because it allows establishing an indirect causative link between habitat loss and stability (especially in the case of contiguous habitat loss for which average interaction strength increases and communities become more unstable). How did the authors convert the various individual-level interactions in the different cells in terms of interaction strength? Maybe I lost the description of this aspect in the text, but for sure more emphasis and explicit description should be dedicated to it. How were the interactions deduced for the taxa starting from individual-level information? Was the strength defined in terms of frequency of interactions between pairs of species? If yes, was the frequency weighted in terms of energy/biomass flowing from resources to consumers? These aspects have consequences on the definition of stability (L407-425) but also influence the quantitative version of many network indices (L395-405 and Table 1 in the manuscript).

The manuscript would benefit of a paragraph in the discussion where possible limits of the modelling framework adopted should be clearly summarized. Such limits refer to some purely methodological aspects listed above (e.g. use of parameters not inferred from empirical or experimental data; consideration of unrealistic conditions to test the role of habitat fragmentation only respect to species richness – see L283-286) but should also honestly present the ecological limits of their application (e.g. the IBM is adapt for the study of terrestrial systems but not suitable for aquatic food webs).

Minor comments

--- L36-37 = How can the authors state their results are generalizable to communities of multiple species? I pointed out possible issues in the application to aquatic systems (in the general comments), which already represent a clear limit to generalization. Also the lack of biological evidence for most of the parameters adopted in the IBM consists of a limit to talk about generalization. I recommend removing this sentence and clearly state the application is particularly suitable to terrestrial food webs.

--- L37-39 = Instead of writing that the causal chains (operating behind responses to two schemes of habitat loss) have been revealed, I would appreciate the explicit description of these two mechanisms. One is the increase of interaction strength that leads to instability (in case of contiguous habitat loss) and another is the impact on diversity and network structure that depend on habitat fragmentation.

--- L38 = While reading the abstract and the introduction, it was not fully clear to me the meaning of “types of habitat loss”. The reader could have immediately a better idea about one key aspect of the work if the authors would state in the abstract that the two types of habitat loss investigated are

random and contiguous. They should use more these terms in the introduction (e.g. see L64-67) as well, since in the version I reviewed they got clear only starting from the methods section. Moreover, the fact the authors refer to the type of habitat loss with different terms might increase confusion (e.g. L64 = "how destruction takes place"; L74 "nature of habitat loss"; L358: "two habitat loss scenarios").

--- L47 = Replace "especial" with "particular"

--- L58 = Remove "s": "empirical data on insect food-webs show that..."

--- In the whole manuscript = I suggest using the hyphen when writing "individual-based model".

--- L169-171 = This part (with citation) does not belong to the results; it should be moved to discussion (eventually).

--- RATP is mentioned for the first time in the text at L186 and its definition is provided in Table 1 only. I suggest including the meaning of RATP the first time is mentioned in the text (and to include it in the text, with the full name of such index).

--- L217-219 = I do not see how the authors can state their findings (i.e. the link between habitat area, interactions strength and stability) are generalizable to communities of multiple species and interaction types. Rather, I highlighted how their application has validity in a specific type of systems (i.e. terrestrial communities) and conclusions cannot be drawn for other interaction types without further analysis (e.g. social interactions), since the inclusion of other interaction types can escape the logic defined by the specific functions codified for the IBM presented here.

--- L390-393 = Did the authors consider quantifying the diversity of the network structure?

--- L414-417 = I would be careful with these statements. Asynchrony in the abundance/biomass fluctuations across trophic groups can also contribute to stabilize ecosystem functioning (e.g. keeping constant biomass production in spite of seasonal fluctuations). Here fluctuations appear to correspond with a negative property (i.e. instability), but such consideration should be attenuated as often fluctuations are the key to maintain constant ecosystem functioning and the provision of services. Some words should be spent by the authors to clarify this aspect.

--- L475 = "plant-pollinator" (remove an extra hyphen).

--- There are too many references, an issue that often translates in blurred messages. One possible way to reduce them is by removing some redundant ones (e.g. at line 84, suggestions 21-27: remove those not used anymore in the text, if any).

--- Caption of Figure 1 (line 6) = Do the authors mean that normalization was obtained by setting the maximum value of each row to 1? If yes, say it explicitly.

--- Caption of Figure 2 (line 1) = "Interaction strength..." (remove "s").

--- Figure 2 = Which is the reason for the differences in the values plotted in panels A and B when habitat loss is 0? I thought this initial point should show equivalent distribution of values for IS and CV_pop (irrespectively of the habitat loss scenario) when no habitat loss occurred yet (i.e. x-axis value = 0). I also suggest using the same range over y-axis for both panels A and B, especially for CV_pop.

--- Table 1 = There are a series of mistakes (or, at least, improper wordings): (1) the terms b.k and b.. refer to biomasses in case of both generality and vulnerability (these are properly defined in Lurgi et

al. 2015, but here the definitions were modified and lost their correct meaning); (2) in nestedness, you should remove the repetition of “of” under the metric column; (3) I am not sure b_{ij} is the number of interactions, as written in the definition of Mean IS (I would expect this to be related to actual biomass flowing from prey i to predator j – this might also help to address my concerns related to the definition of interaction strength); (4) usually CV is expressed as %; (5) not clear the rational reason for focusing on the last 200 simulation steps to calculate CV population and CV range (it should be explicitly mentioned, at least in the text).

--- Appendix S1 needs some text editing (e.g. include only the year “2015” in parenthesis for the citation at page 1, line 2; adjust citation format in the caption of Figure 1 – please double check the whole document to fix this type of issues). Also, in the footnote of the first page it should be “...is also provided in the supporting...”

--- Appendix S1 = At page 6 the reference should be to Appendix S2 and the authors should remove the following words: “Throughout the thesis”.

--- Appendix S2 = The authors stated the criteria that led at the selection of the variables used for the SEM (e.g. RATP, Links, IS). I liked the clear description but do not understand why the number of individuals was excluded (as it shares all of the features displayed by the variables considered in the SEM). An explanation for the exclusion of the variable “number of individuals” is needed.

--- Appendix S2 = The last paragraph refers to MAI ratio that was never defined in the manuscript (it is presented in Lurgi et al. 2015); also, this last paragraph is poorly coordinated with the rest of the Appendix S2. Moreover, which is the meaning of ACV?

--- Some adjustments are needed in Table 1 of Appendix S2 (it should be Table S2): (1) provide references for the argument of HL – Links; (2) rewrite HL to IS as “...interaction strengths is offered by Tylianakis et al. (2008) and Hagen et al. (2012).”; (3) Links to RATP, the citation should be at the end of the definition; (4) Links to CV population and CV range, rewrite as: “...May’s seminal work (1972) is that...”; (5) what stated in the motivation of the link RATP to IS can also be not completely true if we consider the non-random and non-homogeneous distribution of strong and weak links in food webs (see Neutel et al. 2002) – I would be careful with the statement provided and include my thoughts; (6) in the definition of the link from IS to CV range rewrite as “perhaps”; (7) in the link from CV range to CV population remove the repetition for the word “types”.

--- Appendix S3 = There is a weird self-citation to Appendix S3, and the numbering of Figure and Table should consider the name of the Appendix (e.g. Figure S3 and Table S3). Also, in a couple of cases the indication ± 20 misses the symbol “%” and there is a wrong reference to Appendix S1 for SEM (it should be Appendix S2).

--- Supplementary Figures should be renumbered (this applies to all appendices, to avoid having many Figures 1 for example) and the resolution improved.

References

Allesina, S. and Bodini, A., 2004. Who dominates whom in the ecosystem? Energy flow bottlenecks and cascading extinctions. *Journal of Theoretical Biology* 230:351-358.

Berlow, E.L., Neutel, A.M., Cohen, J.E., De Ruiter, P.C., Ebenman, B.O., Emmerson, M., Fox, J.W., Jansen, V.A., Iwan Jones, J., Kokkoris, G.D. and Logofet, D.O., 2004. Interaction strengths in food webs: issues and opportunities. *Journal of Animal Ecology* 73:585-598.

Black, A.J. and McKane, A.J., 2012. Stochastic formulation of ecological models and their applications. *Trends in Ecology & Evolution* 27:337-345.

Bolnick, D.I., Amarasekare, P., Araújo, M.S., Bürger, R., Levine, J.M., Novak, M., Rudolf, V.H., Schreiber, S.J., Urban, M.C. and Vasseur, D.A., 2011. Why intraspecific trait variation matters in community ecology. *Trends in Ecology & Evolution* 26:183-192.

Crooks, K.R. and Soulé, M.E., 1999. Mesopredator release and avifaunal extinctions in a fragmented system. *Nature* 400:563-566.

Giacomini, H.C., De Marco Jr, P. and Petreire Jr, M., 2009. Exploring community assembly through an individual-based model for trophic interactions. *Ecological Modelling* 220:23-39.

Lurgi, M., Montoya, D. and Montoya, J.M., 2016. The effects of space and diversity of interaction types on the stability of complex ecological networks. *Theoretical Ecology* 9:3-13.

Neutel, A.M., Heesterbeek, J.A. and de Ruiter, P.C., 2002. Stability in real food webs: weak links in long loops. *Science* 296:1120-1123.

Van Tussenbroek, B.I., Villamil, N., Márquez-Guzmán, J., Wong, R., Monroy-Velázquez, L.V. and Solis-Weiss, V., 2016. Experimental evidence of pollination in marine flowers by invertebrate fauna. *Nature Communications* 7:12980.

Response to reviewer's

We would like to thank the three reviewers for their very insightful comments and suggestions, which we believe have substantially improved the manuscript.

Reviewer #1 (Remarks to the Author):

Review of "Routes to instability under habitat loss: an investigation of multitrophic communities" by McWilliams et al.

I reviewed a previous version of this manuscript for a different journal, and provided a number of comments and suggestions to improve the manuscript (included below). I am disappointed to find that none of my suggestions have been incorporated into the current version. In fact, the only difference I can find in the current version that the Results section has been moved before the Methods section; all text appears again verbatim. (I was not able to see if changes had been made before accepting to review the current version.)

I spent a good amount of time reading and thinking about the manuscript in the hopes of improving both the science and its presentation, and encourage the authors to consider at least some of my suggestions.

Although previously I was supportive of publication, the lack of desire of the authors to make even the simplest of changes to improve the manuscript leaves me decidedly neutral about this work. (For example, in my previous review I noted that the authors had mistakenly included two consecutive "of"s in Table 1 in the row about "nestedness." I see that this error is still present.) I have said my piece in the previous review and offer those comments to help with the decision-making process at Nature Communications.

We deeply apologise for this oversight. We appreciate very much the reviewer's comments and suggestions, which we have now considered very seriously in order to develop a revised version of the manuscript. We hope that the changes made are consistent with the reviewer's expectations.

""

[Previous review of the present manuscript]

Review of "Routes to instability under habitat loss: an investigation of multitrophic communities" by McWilliams et al.

The authors propose an individual based model of community dynamics to investigate how habitat destruction affects the stability of ecological communities comprising multiple interaction types. They consider two types of spatially-explicit habitat destruction: successive loss of random and contiguous habitat patches; and two types of interaction: trophic and mutualistic interactions. They assess stability in two ways: temporal variation in species population abundances and temporal variation in species range area. The authors use their modelling approach to show that communities are more affected by random habitat loss, and that corresponding changes in stability are insensitive to the fraction of mutualistic interactions in the community.

The manuscript is clear and well-written, and addresses an important question in community ecology: does modelling multiple rather than single interaction types change how we expect

communities to respond to habitat loss, and will responses vary depending on the form of habitat loss? The authors propose an interesting modelling approach to begin exploring this and related questions. I have three major comments about the work: 1. Can we definitively rule out the counterclaim that mutualism does have a qualitative effect on community stability? 2. The modelling approach should be expanded to allow for a more general and flexible investigation of community responses to habitat loss. 3. Some language should be clarified to avoid confusion. If these comments could be adequately addressed then I would be supportive of publication in [JOURNAL NAME].

1. Can we definitively rule out the counterclaim that mutualism does have a qualitative effect on community stability?

The authors build ecological networks with multiple interaction types in two steps (P8L31): first, they build a food web using the niche model, then they replace a fraction of the trophic interactions between the two lowest trophic levels by mutualistic interactions. The authors then vary the fraction of *replaced* interactions between 0% and 100%.

Before I accept the authors' claim (P3L18) that results on stability are insensitive to the fraction of mutualistic interactions in the community, I have two questions:

At 100% replaced interactions, what is the *total* fraction of mutualistic interactions to trophic interactions?

On average, at 100% of replaced interactions the total fraction of mutualistic interactions in the food web is 23%.

Presumably their communities have more than two trophic levels, so even with 100% replaced interactions the overall fraction of mutualism in the community could still be very small. In which case, it would be unsurprising that a small number of mutualistic interactions had little effect on community stability.

What happens if interaction types are distributed randomly throughout the network, or model networks are based on distributions of interactions based on empirical communities with multiple interaction types?

Given that this work is largely a theoretical contribution, I would like to see a more comprehensive study of possible cases (even if some of the results end up as supplementary information).

The referee raises a valid point here regarding how the fraction of mutualistic interactions was modelled. Mutualistic interactions in our model can only occur between resources and primary consumers, and the fraction of mutualism is changed by varying the number of mutualistic to trophic interactions between the basal and the 2nd trophic level (as opposed to a random distribution of interactions, for example as in Mougi & Kondoh 2012). Although mutualistic interactions can occur at other positions in the network of interactions (e.g. ant-aphid interactions, both within the 2nd trophic level), we based this modelling decision on the fact that the most studied mutualistic interactions occur between plants and animals (e.g. seed-dispersal, plant-pollinator interactions).

The reviewer correctly asserts that a 100% of mutualistic interactions in our model does not mean all interactions in the community are mutualistic. Indeed, the highest fraction of mutualistic interactions in the community when all trophic interactions are replaced by mutualistic ones between basal resources and primary consumers is 23 %

on average. With such fraction of mutualism, we cannot conclude that the influence of mutualism on community stability is qualitatively neutral or irrelevant. However, we doubt that communities with a 100% of mutualistic interactions across all trophic levels exist in nature. In fact, replacing all interactions in our modelled food webs by mutualistic ones would result in unrealistic ecological networks, since we are using as a baseline food webs constructed using the niche model. This model was chosen because it ensures the generation of networks that resemble food webs in several features. Mutualistic networks on the other hand, have been proven to display structural features very different from those of food webs. Creating fully mutualistic networks by changing all the links to mutualistic ones would thus generate meaningless networks because it would enforce a food web structure over a network of exclusively mutualistic interactions, this leading to results unable to inform about the response of realistic communities to habitat loss.

We nonetheless thank the reviewer for pointing out this issue and have made substantial changes in the manuscript to account for it. In the revised manuscript, we are more explicit about how mutualism is calculated (lines 480-488), how it differs from a random distribution of interactions (lines 367-370), and do not rule out the influence of mutualism on community stability (lines 357-374).

Finally, the authors consider trophic (+-) and mutualistic (++) interaction types. For completeness, it would be interesting to also consider competition (--).

Competition for space is actually in place in our model given that cells are only available to a limited number of individuals in our simulated landscape. In addition, exploitative competition also arises as individuals can share the same resources. Hence, competition is implicit in our model which thus covers the three types of interactions mentioned by the referee. Practically, we focused on the explicit modelling of trophic and mutualistic interactions to contribute to bridge the gap between food webs and mutualistic networks (a historical dichotomy). This approach makes our results comparable with studies that have recently started to combine trophic and mutualistic interactions. We have specified this in the revised manuscript (lines 393-398).

2. The modelling approach should be expanded to allow for a more general and flexible investigation of community responses to habitat loss

The authors consider two versions of habitat loss: random and contiguous. It would be useful to present a more general model that can describe a wider range of possible habitat loss scenarios. For example, the authors could have a single parameter (h) that controls how habitat patches are selected for destruction at each time step. Values for h would range from perfect anti-correlation of selected patches ($h = -1$), to slight anti-correlation ($h = -0.5$), to random ($h = 0$, one of the two currently studied cases), to slight correlation ($h = 0.5$), to perfect correlation ($h = 1$, the current, contiguous, case).

Analyses could then explore how community responses to habitat loss vary along a continuum. This would represent a more helpful theoretical baseline than the simple dichotomy of random and contiguous habitat loss. It would also permit analysis of how patch-size distributions are related to community responses.

We have thought a lot about this fundamental point raised by the reviewer. To address the reviewer's concern, we decided to: (1) provide a better justification for the scenarios used, and (2) incorporate an intermediate scenario.

Regarding the first point, we believe that we have not been clear enough in the manuscript about the reasons to justify our choice of random and contiguous scenarios of habitat loss. The reference context of this study lies within the land-sparing vs land-sharing framework. These are the main approaches proposed in the literature for optimising land use and biodiversity conservation: while under land sparing, biodiversity is essentially concentrated into one or a few large habitat fragments; under land sharing, it is distributed across the whole of a landscape but in a large number of smaller, fragmented, patches of habitat. These two approaches therefore lie at opposite ends of a continuum, and they have motivated our initial choice of habitat loss scenarios; that is, random habitat loss corresponds to land-sharing whereas contiguous habitat loss represents land-sparing. This choice of scenarios also makes our results comparable with the majority of studies on habitat loss, which have modelled either random (Namba et al 1999; Szwabinski & Pekalski 2006; Jager et al 2006; Travis 2003), or contiguous habitat loss (Hiebeler 2000, 2004; Hiebeler & Morin 2007), or both (Ovaskainen 2002; Alados et al 2009; Soares dos Santos 2010). Besides, we believe that considering a continuum of habitat loss scenarios would dilute the essential message of our work, that is, the effects of habitat loss on interaction strengths and community stability depend on how the habitat is lost, and hence, on the general approach followed for optimising land use.

Despite this, and considering our second point above, we agree with the reviewer that including intermediate scenarios of habitat loss would be very relevant from a theoretical perspective, for example to investigate the transition from one regime to the other. In the revised manuscript, we add a new scenario of habitat loss that corresponds to a degree of spatial correlation of 0.5 (halfway between random and contiguous loss), and analyse the response patterns of the simulated communities. For this intermediate scenario, we find that diversity, network properties, and community stability show an intermediate response relative to random and contiguous habitat loss. A newly-developed redundancy analysis also supports this, showing that community responses across habitat loss scenarios are gradual, going from random through intermediate and ending in contiguous loss (new figure 2). Moreover, the relationships between habitat loss, the strength of species interactions and the temporal variability of species abundances, are also consistent with this intermediate pattern of change (Figure 3). Finally, mobility analysis for the intermediate scenario shows that the mobility of individuals across the landscape is intermediate relative to random and contiguous habitat loss (Figure 5). Collectively, the results for the three scenarios suggest that the magnitude of changes experienced by communities due to habitat loss are consistent with a gradual transition, and can be predicted based on degree of spatial autocorrelation of lost habitat. The nature of our simulations, which keep track of thousands of individuals in a spatially-explicit context, and with unique bio-energetic values, mobility outcomes and interactions, makes the exploration of a continuous gradient of habitat loss scenarios unfeasible. However, we believe that the gradual nature of the transition observed across the three scenarios of habitat loss makes it unnecessary to conduct an investigation of a wider range of intermediate habitat loss scenarios.

We have made substantial changes in the revised manuscript to justify more explicitly the choice of the two scenarios of habitat loss (e.g. lines 131-136, 531-536). We have added new material related to the intermediate scenario and discuss the implications in a broader context (e.g. lines 139-142, 159-160, 169-171, 180-187, 199-202, 234-235, 247-249, 256-258, 269-271, 327-345, 385-393, 541-543, 614-626). We thank the reviewer for his/her comment as we believe that these additional analyses and clarifications have greatly improved the manuscript.

3. Some language should be clarified to avoid confusion

The authors use the term "stability" throughout the manuscript, but the term is used to refer to very different concepts. This is unavoidable because researchers have used many different technical approaches to study "stability" (as mentioned, P6L26). To minimise confusion, I suggest the authors define how *they* measure "stability" earlier in the manuscript. This could be done in the Abstract (P3L16), and in the paragraph introducing their work (P7L17). The authors should also be careful not to make comparisons between results for different definitions of stability (P20L21, P21L22) without explicitly mentioning which definitions are being used.

We have specified in several parts of the manuscript how stability is defined. Comparisons between our results and other published results on stability explicit mention the definitions of stability given in those previous works (e.g. lines 31, 92, 118-128, 324-325, 361-364).

Given the prominence of "stability" in the manuscript, the authors should consider, perhaps in the Discussion, how results for *their* definition of stability fits in with previous or future work that addresses a similar question but with a different measure of community "stability".

We have not found studies looking at how habitat loss affects stability of biological communities, aside from those measuring stability as species / community persistence. Theoretical studies on the relationship between network architecture and stability of undisturbed communities with trophic and mutualistic interactions often define stability as the proportion of stable communities following May's stability criterion – i.e. using asymptotic resilience (e.g. Allesina & Tang, 2012; Mougi & Kondoh, 2012; Sauve et al, 2014). Although these works refer to undisturbed communities (e.g., communities not subject to the loss of habitat), we relate our results to such studies in the revised manuscript (lines 361-367).

The authors do not demonstrate any kind of causality in the study, so I suggest avoiding the use of this term. Even within the context of their models (for which establishing causality should be more straightforward than in general) they do not formally establish a causal chain (by comparing the relative importance of directly modelled processes, for example). Although I value structural equation models, personally, I do not consider their outputs as sufficient evidence to imply causality (i.e., how can we be sure whether it is a loss of interactions that causes a change in relative abundance, or a change in relative abundance that causes a loss of interactions). Some points where statements about causality should be reconsidered include: P3L25, P3L32, P17L45, P18L37, Figure 3 caption.

This is actually an open debate in the current literature. But we agree with the reviewer, and we have modified the text throughout the manuscript to avoid explicit mentions to causality.

The authors should clearly and unambiguously define "interaction frequencies" and "interaction strengths" (P11L35). This would serve to avoid any confusion between concepts including: i. the number of recorded contacts between individuals of two given species in a model or empirical data, ii. interaction parameters in a bioenergetic model, and iii. summary statistics of interaction distributions (e.g., P11L44 and P11L47).

Following the reviewer's suggestion, we have defined and differentiated interaction frequencies and interaction strengths to avoid confusion (lines 580-586).

Minor comments

P3L28. The "two types of habitat loss" are not introduced in the abstract.

The types of habitat loss are now introduced in the abstract.

P7L8. This sentence needs clarifying: are the authors trying to say it is not clear how habitat loss affects the *relationship* between interaction patterns and stability?

We have clarified the sentence following the reviewer's suggestion.

P7L54. The authors could also cite examples of mutualistic interactions *destablising* ecological communities, e.g., Staniczenko et al. (2013) The ghost of nestedness in ecological networks. Nature Communications, 4, 1931

The cited work has been included as evidence that purely mutualistic interactions can destabilise communities. We also cite this work to show that, despite our study does not find support for a (de)-stabilising effect of mutualism, this may result from the fact that our communities have a lower fraction of mutualistic interactions (as opposed to, for example, Mougi & Kondoh 2012) (lines 148-151, 367-374).

P8L45. "...replacing some herbivorous interactions..."

Done.

P9L42. There needs to be a paragraph here explaining how species are initially distributed across the landscape (and how many individuals per species and the "shape" of the landscape and how boundary effects are handled); some of the text from P10L32 could be moved earlier.

Done.

P10L35. How do mobility values vary between individuals and species? Do individuals perform a random walk?

We explain this in the supplementary material: if the individual is a plant it does not move. Otherwise a neighbouring cell is selected uniformly at random. If the selected cell contains a non-basal species, an interaction (feeding) occurs according to the rules of interaction. Otherwise, if there is available space in the selected cell, the individual moves there. The motion is therefore a two-dimensional random walk, as represented in figure 1 of the supplementary methods 1:

Figure: The trajectories of two individuals over 12 time steps are shown in black and dark grey. The distance-1 neighbourhoods of the two individuals on the first-time step are shown in light grey.

P13L49. Although I like the headings in the Results, should they match the three categories introduced on P11?

We have changed the first headings in the results to match those in the methods section.

P14L3. Are the authors meaning to suggest that random habitat loss is a good thing? Or do they mean that communities were less unstable under random habitat loss compared to contiguous habitat loss (although this contradicts the previous sentence)?

Communities under random habitat loss appear to be less variable over time, but this does not mean that random habitat loss is positive for biological communities. Notably, when looking at other aspects of biodiversity and community structure, random habitat loss results in communities that are structurally simpler and more fragmented or spatially disconnected. Such communities experience important changes, such as the collapse of predator populations, which has been associated with a loss of overall variability. We include additional text in the discussion to clarify this (lines 320-325).

P14L26. Are there identifiable patterns with contiguous habitat loss?

Some network properties change across the gradient of habitat loss in the contiguous scenario, but this occurs only at certain fractions of mutualism, and the effect sizes are small (e.g. supplementary figures S2, S3). In general, contiguous loss compresses communities to smaller regions of space where populations are destabilised (i.e., population dynamics experience more variability) without major changes in their network structure.

P14L31. Clarify how links are lost: is it due to no co-occurrence between species in a grid cell, or a *shift* away from an interaction even though the species do co-occur?

Done.

P16L3. Mention that results in this paragraph are from linear models.

Done.

P16L40. Explain numbers like "0.71".

Done.

P16L51. How exactly does this statement relate to the results just presented? Explain *how* the distributions of interaction strengths are changing, and how this affects stability in the two habitat loss scenarios.

Done. We have also included an additional figure in the supplementary material (supplementary figures, figure S5).

P17L17. Clarify this sentence.

Done.

P17L51. What is "the link between habitat area, interactions [sic] strengths and stability" exactly?

We have modified this sentence.

P18L10. It would be helpful if the definition of stability in terms of variability in population abundances was restated.

Done.

Table 1. In "nestedness", there are two "of"s.

Corrected.

Table 2. Are all values significant, and, if so, at what level?

Done.

Figure 1. What do the numbers on the scale mean?

The numbers on the scale indicate the size of trend detected. Trend size is normalized within each row, such that the colour indicates the effect size for each variable relative to different fractions of mutualism. We clarified the meaning of the scale in the legend.

Figure 2. What about CV_range?

CV range is included as supplementary figure S4. In figure 2 (now figure 3) we wanted to highlight the relationship between habitat loss, the strength of interactions and the variability of population abundances.

Reviewer #2 (Remarks to the Author):

McWilliams et al present an interesting simulation on the effect of habitat loss on community stability. The work presented here is based on extensive previous knowledge and a well-developed individual based model yet it presents a novel and timely extension by including the effect of habitat loss and including two types of interactions. The results are in general well discussed and presented, although at some points the text is difficult to follow. However, although I believe the work is a great contribution I have several concerns regarding the applicability and usefulness of these approaches to real-world settings and also am afraid that the contents of these paper might be better suited for a more specialised journal.

The authors suggest in their model that once a cell is destroyed during the habitat loss phase then no species can move there and no offspring can be placed there. I understand this is a simplified model of reality but wonder if it wouldn't be more realistic if some species were allowed to use these destroyed cells, given that many species are able to live in the matrix of newly fragmented habitats. I wonder whether not allowing for this has an important effect for your results. Also, there is no consideration of edge effects in the model and how they could affect the community dynamics. In particular, given that one of the interactions considered is the mutualistic interaction of seed dispersal, edge effects are particularly important as in many cases for example more fruits are produced in edge habitats.

The reviewer is right in that certain species can moderately thrive in the matrix of disturbed habitat. However, the aim of our study is to investigate the effects of intensively-transformed/managed habitats – meaning the absolute loss of habitat for any species – on the structure and stability of biological communities. A useful analogy are agricultural systems, which comprise a mosaic of different types of habitat fragments (e.g. crop land, (semi)-natural habitat). Certain agricultural exploitations allow some biodiversity to inhabit and thrive on crop land, for example organic farming. Conversely, intensively-managed agricultural systems deliberately prevent the establishment of wild species on crop land by using chemical and mechanical methods, thus restricting biodiversity to fragments of (semi)-natural habitat. In fact, this drives the observed worldwide decline of pollinator diversity in intensively-managed agricultural systems, mainly due to the increase in crop land at the expense of the (semi)-natural habitat where wild pollinators nest and take refuge. In the revised manuscript, we specify that our model refers to intensively-managed landscapes, and that for different management scenarios that allow matrix habitat to host some levels of biodiversity, the effects of habitat loss on communities may be different from the ones reported here (lines 398-403, 544-545).

Concerning edge effects, we identify two classes of potential edge effects in our model. The first one is related to how boundary conditions are dealt with in the simulated landscape. In this case, edge effects are prevented by using periodic boundary conditions, i.e., the topology of the simulated landscape is toroidal (lines 570-573). The second type of edge effects relates to the preference of certain species for habitat edges. For example, edge habitat has been observed in some cases to be preferred by seed-dispersing animals (Restrepo et al, 1999), which may explain why more fruits are produced at the edge of habitats, or early-successional species (Imbeau et al 2003). However, we could not find evidence to conclude that such edge effect systematically affects all types of interaction and species. Further, empirical evidence shows that some species prefer core or interior habitats and display patterns of edge avoidance (e.g. Zurita et al, 2012). Therefore, our model assumes all species are initially neutral to edge or core habitat. Despite this, and even though including species preferences for either edge or core habitat is beyond the scope of this work, edge effects are indeed expected to emerge in our simulations given the way mutualistic interactions, and what follows after the actual interaction happens, are specified in our model. For example, it

is more likely that mutualistic individuals, and the plants they disperse, will move to an empty, non-destroyed cell at the edge of a plant cluster; therefore, plants closer to the border of a given patch of plants have a higher probability of being dispersed. This results in a sort of 'more fruits', or at least 'more dispersed fruits' scenario, that is more likely to emerge at the edge of patches.

The revised manuscript includes a paragraph with the limitations of this study; two such limitations refer to the two issues raised by the reviewer: (i) the possibility of some species to inhabit fragmented cells in other type of management scenarios (lines 398-403), and (ii) the absence of species preferences for edge or core habitat (lines 403-410).

The authors present the two scenarios of habitat loss as the two extremes that we might expect in nature, yet they are not equally common, rather the contiguous scenarios tend to be the norm while the random one would be something rare. Even in fragmented areas where habitat loss is not contiguous, it is also not random. The authors could comment their results in light of this difference between the probabilities of scenarios taking place. Also, the authors could place these two scenarios within the available literature, whether it's the fragmentation or the land sparing-land sharing or whichever they choose. At the moment, the reason why these two contrasting scenarios were chosen and what their implications are is diluted throughout the text.

We thank the reviewer for his/her suggestion, which has been also raised by reviewer #1. The context of this study and the reason for choosing the two habitat loss scenarios (random and contiguous) is indeed based on the land-sharing vs land-sparing dichotomy. Land-sharing and land-sparing lie at opposite ends of a continuum: while random habitat loss corresponds with land-sharing; land-sparing is better represented by contiguous habitat loss. This choice of scenarios also makes our results comparable with the majority of studies of habitat loss, which have modelled either random (Namba et al 1999; Szwabinski & Pekalski 2006; Jager et al 2006; Travis 2003), or contiguous habitat loss (Hiebeler 2000, 2004; Hiebeler & Morin 2007), or both (Ovaskainen 2002; Alados et al 2009; Soares dos Santos 2010). In the revised manuscript, even though we have added an intermediate habitat loss scenario, we are more explicit justifying the choice of random and contiguous loss, and discuss the implications of our results within the land-sharing vs land-sparing framework (lines 133-139, 327-345, 531-536).

The text is difficult to follow sometimes as the acronyms used and many of the concepts are not described before they are used the 1st time. For example, it is unclear what MAI means but it seems it might be related to FM, fraction of mutualists? In line 126 you start to use HL for habitat loss but you have never introduced it before and the reader has to assume that is the case. Then you indistinctly use "habitat loss" or "HL" and "IS" or "interaction strength", stick to one of the two and be consistent.

The abstract by itself is very difficult to follow. For example, it is not clear when your first read it what the authors mean by "hybrid communities". Neither is it clear what they mean here by "two types of habitat loss". I understand that given word constraints this is not always an easy task but the abstract alone should be able to tell the reader what the paper is about and what the general conclusions are.

We have addressed these issues in the revised manuscript in order to increase clarity.

In the Introduction L63-74, the authors describe habitat loss and its effects, yet they tend to mix concepts. First, they are comparing random or spatially-correlated habitat loss and then they introduce habitat fragmentation, which entails habitat loss but also many other effects (edge effects, loss of connectivity), which this paper in particular does not take into account.

We have changed this paragraph and removed the parts that relate to edge effects and connectivity.

Across the whole paper and in the Abstract and Intro sections the authors claim that their study addresses how communities “respond to habitat loss prior to species extinctions”. However, the number of species does not remain constant throughout the simulation, but rather some species go locally extinct (as shown in negative effects of habitat loss on number of species on Fig. 1), so I am not sure this is correct. Maybe I have not understood properly, as the authors suggest this is countered by high immigration rates. But at present I believe this needs clarification.

We understand the reviewer’s confusion. In figure 1, the trend size is normalized within each row, such that the colour indicates the effect size for each response variable relative to different fractions of mutualism, but does not provide any information about the actual effect size. Indeed, the largest effect size detected in all model simulations corresponds to a loss of ~0.5 species across the full gradient of habitat loss. It is for this reason that we conclude that extinctions are not observed. We clarify this issue in figure 1 of the revised manuscript.

The authors introduce their hypothesis at the end of the Intro section yet some of their expectations are not very elaborated. For example, why do the authors expect that the fraction of mutualism will affect stability? What kind of negative effects are expected from Hypothesis 2?

We have elaborated more the hypotheses of this study based on previous works (lines 117-151)

Sometimes the text is hard to follow. For example:

L137-140: “The contrasted responses of network...suggesting an important difference...” A difference compared to what?

L140: “For some of these properties...” Which properties?

L169: here you are referring to the results of a previous study, not the one presented here. Please specify this.

L171: Similar how?

L186: Explain what SEMs are, it’s the 1st time you mention them.

L191: which is the latter case? This sentence is unclear

You use habitat destruction and habitat loss interchangeably, please stick to one terminology consistently.

The text has been clarified / modified in all of the places highlighted above by the reviewer.

L208-243 The first part of the discussion basically repeats the results already enumerated in the results section with no implications or explanations to the observed patterns.

We believe that a paragraph summarizing our results provides a necessary frame for the discussion that follows.

L254: the authors suggest that because the abundance of top predators remains constant under the contiguous habitat loss scenario, interaction strengths increase and this results in greater variability. However, they have not included home ranges for any of these top predators, and the decrease in habitat available might mean some might go extinct because their home range exceeds this habitat. Also, as habitat available decreases, competition between species might increase, yet this has not been taken into account either. A list of caveats would be a good way to proceed.

We agree with the reviewer's comment and have added a paragraph in the discussion describing the caveats of this study (lines 385-421).

L253: "...increased abundance variability in forest fragments" you are discussing the findings of the contiguous habitat loss scenario in the light of the fragmentation literature, yet fragmentation would be more in the lines of the random scenario in your case.

We have removed this part from the paragraph.

L260-261: please re-write this sentence, it makes no sense.

This sentence has been removed because the second sentence introduces well the paragraph and it is thus not necessary.

L262-264: that's because you assume the matrix is absolutely unsuitable even for movement between habitat areas, which is not realistic in many cases.

Please see previous comments above. The new limitations paragraph includes material on this issue.

It seems like the Methods section originally followed the Introduction and was then shifted because reading the Methods at the end explains many things that are not clear as the paper is presented right now. Consider changing this.

We have made changes to increase the clarity of the revised manuscript.

Reviewer #3 (Remarks to the Author):

Manuscript #: NCOMMS-18-15210-T

Title: Routes to instability under habitat loss: an investigation of multitrophic communities

General comments

In this manuscript, McWilliams and coauthors present an individual-based model to study the effect of habitat loss on multitrophic ecological communities. Main novelties are the inclusion of mutualistic interactions and the evaluation of changes induced by habitat loss prior species extinction occurs. The analysis of simulation outcomes highlighted that two extreme types of habitat loss dynamics have fundamental differences on ecological communities. When habitat loss happens according to the contiguous scheme the food webs get more unstable, and this response is strongly driven by increase in the average interaction strength. If the habitat loss follows the random scheme the diversity and the structure of the community are highly impacted due to landscape fragmentation. Contrarily to expectations, the proportion of mutualistic and antagonistic interactions has almost no consequences on the patterns identified by the study. In what follows there are a series of general, more substantial concerns that should be addressed by the authors. General comments refer to the individual-based model (IBM), the structure of the introduction, the meaning of stability, and the lack of a paragraph to properly discuss the limits of the simulation approach. Other minor comments offer both methodological suggestions and editing hints.

The manuscript is well written and I appreciated the effort done by the authors to condensate in simple messages the findings of their analysis. It is well known that the appeal of IBM in ecology has often been impaired by difficulties in obtaining results of general value (i.e. IBM are often case-specific but a few exceptions exist in the literature; e.g. see Giacomini et al. 2009) and also the need of highly parameterized equations often hindered their success or caused skepticism (i.e. modelers risk to address complexity with other complexity, failing to capture key mechanisms governing systems' dynamics; e.g. see Black and McKane 2012). I believe these issues have been partially addressed by the present manuscript. The use of SEM to interpret under a cause-effect perspective the results of simulations goes in the direction of simplifying the complexity and illustrates possible general mechanisms behind the recorded responses. However, there are some concerns related to the structure of the model (type of processes simulated, equations used and parameters chosen) and its generality (i.e. it is suitable for terrestrial systems but not aquatic ones) that should be addressed prior to consider the manuscript for publication.

More efforts should be dedicated to explain the value of such approach in the context of terrestrial and aquatic ecology. My feeling is that while the application of this IBM can be appropriate for the case of terrestrial ecology, some concerns exist in the case of marine systems. For example, how to explain the mutualism in marine systems in an analogous way as it is presented in the manuscript for terrestrial plants? There are sure mutualistic relations in marine systems, but I can hardly envision their behavior in the sense coded by the MUT_EFFICIENCY function. Although an example of pollination in the sea has been recently described (see Van Tussenbroek et al. 2016), this is far from being representative for the marine systems and cannot be used as justification to extend the validity of the IBM to marine food webs. Another reason to define the current framework as inappropriate to model aquatic systems is represented by herbivory (HERB_FRACTION function). In the model the non-mutualistic herbivore consumes a fraction of plant energy and both individuals continue living; this is clearly not the case in most of aquatic interactions involving the planktonic community. For example, in the copepod-phytoplankton interaction the feeding relationship ends with the removal of the phytoplankton cell. What I mean is that this work seems to apply better to terrestrial systems than to aquatic ones, especially given the structure and principles

governing some functions. I suggest changing the emphasis of the manuscript and presenting it as an application that can (eventually) be generalized in the context of terrestrial food webs.

The reviewer is right. Our model and results mostly refer to terrestrial communities. The type of disturbance analysed also concentrates on a major global change component affecting terrestrial systems (e.g. agricultural extension combined with the intensification of cultural practices, urbanisation). We have made this clear in the revised manuscript and add material in the discussion explicitly stating these results cannot be extrapolated to aquatic communities (lines 29-30, 273-275, 398-403, 421, 466-468).

In order to help clarify the issues raised by the reviewer, we have also included additional information about the model in the methods section of the main text and in the supplementary methods 1.

Other specific aspects that are not convincing about the IBM (or questions that should be addressed) are provided in the list of below:

--- Can the authors briefly explain the benefit of using IBM for investigating changes in the properties of multitrophic communities in presence of various habitat loss scenarios? Why did they choose to model the consequences of various types of habitat fragmentation on food web stability using IBM and not ODEs? Are they dealing with small population sizes? Why do authors expect stochasticity to play an important role in the processes under investigation?

To investigate the consequences of habitat loss on multitrophic communities, we needed a spatially explicit framework in which modelling movement at the individual scale is essential. We further defend our choice of an IBM over ODEs in the methods section (lines 449-464).

Our communities do not display small population sizes, except at high levels of habitat loss (we have added a new figure showing this: supplementary figure S1).

We do not claim in the manuscript that stochasticity has a fundamental role in the processes under investigation, but rather add biological realism to our model. Stochasticity is known to affect processes at the individual level, such as the interaction between individuals of the same species and their demographic outcomes, the movement of individuals and the process of immigration, and we wanted our model to include part of this complexity. For instance, the matrix of interactions defined by the niche model determines the species that can potentially interact with each other. However, based on empirical observations, two potentially interacting species do not always do so even if they coexist in space – e.g. plants and pollinators, predators and prey – and this may be due to a variety of factors that are modelled here as a random process. We include this in our model: interactions are realized only if individuals of those two species coexist in space (i.e. meet in the same cells), and given a certain probability (e.g. capture probability in the case of predator interactions). Essentially, this means that two individuals may interact at a certain time and location, but may not do so at a different time or location, thus introducing realism in our model. In sum, our model includes stochasticity to add biological realism, but we do not explicitly state that it is a key factor determining the results.

--- The authors kept the number of species constant by having high immigration rates. To quantify the bias imposed by such (often unrealistic) condition, I suggest performing other simulations for comparison with current results. I am not asking the authors to completely change the structure of their manuscript, but recommend including this additional analysis as new Appendix file. Results could be used to comment the possible consequences of such change

in the IBM (e.g. in a new paragraph that discusses limits of current IBM, and maybe to integrate the sentences at L283-289).

Following the reviewer's suggestion, we have run new simulations with lower immigration rates. In general, although more extinctions occur when immigration rate is lower, the effects of habitat loss on interaction strengths and the temporal variability of population abundances is qualitatively similar. We mention these results in the main text (lines 348-353, 509-511) by reference to a new appendix file (Supplementary Methods 4).

--- Is it realistic to assume that all landscape cells have the same quality? Are there other individual-based models that modulate migration in a landscape depending on the quality of various patches?

The simulated landscape hosts two types of cells: destroyed and non-destroyed. We investigate the effects of intensively-transformed habitats – meaning the absolute loss of habitat for any species – on the structure and stability of biological communities, and therefore biodiversity does not thrive in destroyed cells (see our response to reviewer #2 on this similar topic). For non-destroyed cells, no differences in terms of habitat quality were explicitly included for simplicity (line 519-520), as it would make results difficult to interpret given the potentially confounding factors of habitat quality and loss. However, because of the individual-based nature of our model, cells differ in terms of species as a result of biotic filtering.

There are some metapopulation models investigating individual movement in habitats with different quality, where habitat quality is defined by its temperature (e.g. Jacob et al 2018). However, we do not know of individual-based models that modulate migration in the landscape as a function of habitat quality. What is definitely missing in the literature is migration in the landscape depending on the quality for communities comprising many species.

--- Did all taxa display the same migration rates (L350-353)? Or were migration rates randomly assigned even within each taxon? It would be useful to explicitly read whether intra-specific variability was considered as this might have profound consequences on the outcomes of the model (e.g. see Bolnick et al. 2011).

We used the same immigration rate for all taxa. First, an immigrant individual is created with probability given by the immigration rate. The immigrant species is selected at random from the original species pool. There must be space in the cell for the immigrant to be placed, or the immigrant must be able to feed upon the species present in the cell. Otherwise the immigrant is discarded. Thus, although all individuals of all types bear the same immigration potential, effective immigration does vary across cells. This translates both into intra- and inter-specific variability in local immigration rates. We have added text in the discussion about this (lines 403-410, see also Supplementary Methods 1) and cite the reference suggested by the reviewer.

--- L323 = I would be careful in using the term “hybrid” here. In the framework of modelling literature “hybrid” refers to models that include both deterministic and stochastic components while here the term deals with mixed types of ecological interactions. The term should be either removed/replaced or it should be clearly stated this does not describe a type of model construction/methodology.

We now clearly state that we use the term ‘hybrid’ not as a description of a modelling framework (models with both deterministic and stochastic components), but as a

definition of communities in both trophic and mutualistic interactions, as it is done in the multiplex network literature (lines 443-446).

--- L343 = I would have expected the extinction rates of various species are stochastic (i.e. not defined with a static parameter but stochastically regulated by the execution of simulation). Am I wrong? Or maybe the authors simply refer to stochastic as dependent on a pre-defined probability? Clarification is needed.

The model does not specify species' extinction rates. Rather, individuals die off if their energy levels go below some threshold. This means that individual extinction is totally deterministic and depends on the history of the individual (how much resource it has managed to collect during its lifetime). Extinction 'rates' thus emerge from the model as a system-level property (or more specifically, species-level) out of processes occurring at the individual level. We thus refer to stochastic as dependent on a pre-defined probability. We have clarified this (lines 494-496).

--- L345 = Are demographic processes modelled as stochastic? More generally, how is stochasticity generated (i.e. using deterministic pseudorandom number generation – e.g. analogous to what can be obtained using the `set.seed` command in R – with Monte Carlo method or a variety of it – e.g. Gillespie's Stochastic Simulation Algorithm)? The way stochasticity is generated should be explicitly declared.

All stochastic processes are modelled based on pre-defined probabilities (see last comment), and these probabilities at every time step are obtained using a pseudorandom number generator provided by Python. We have clarified this in the revised manuscript (lines 500-501).

--- L366 = What does one step correspond to (i.e. one hour, one week, one month or one year)? Is it realistic assuming that all taxa have same immigration probability (see Table 1 in Appendix S1) and move with the same velocity (i.e. one cell per step of simulation, if the recipient cell is not occupied by other consumers or excluded from the simulations due to habitat loss)?

We do not specify the length of time steps in the model. Time steps can be as long as required for ecological processes such as individual encounters and movement to happen. The model allows consumers to always take resources, and this is a realistic assumption given that all individuals spend energy in each time step according to bioenergetic rules. This assumption of the model implies that the length of a time step can be seen as the time at which an individual spends a sufficient amount of energy as to feel the need to find more resources. Despite this, we do not specifically assign any time units on purpose, and this is because our goal is not to represent faithfully a particular terrestrial community or specific ecosystem, but to investigate the response of a standard / ideal community to a loss of habitat. The definition of time or the time scale implemented differs across different systems and would make parameterisation more complex. For similar reasons, units of area are not defined either (see response to a comment below). Spatial and time scales are related, and therefore defining time units without a specific spatial scale would be too speculative. This also influences our choice of not defining a particular time scale. We have specified these aspects of the model in the revised manuscript (lines 466-475).

For simplicity, intra- and inter-specific differences in immigration are not considered; although the outcome of immigration does vary among individuals. We have been more explicit about this and added text to the limitations paragraph (lines 406-410).

--- L397 = Which is the rationale behind characterizing network structure during the final 200 steps? Why not the last 100 or 1000 steps? Were simulations exposed to transient phase during

the progressive habitat loss and the authors wanted to get rid of this effect? Were the last 200 steps considered to be stable and with no habitat loss occurring? More details need to be provided in the manuscript.

It is to avoid misleading results due to transient dynamics. We considered the last 200 time steps to be stable and a large enough number to perform statistical analyses. This number is arbitrary, and, as the reviewer states, considering the last 100 or 1000 steps is equally valid if the above-mentioned conditions are met. In other words, results will be robust regardless of the time window used as long as transient dynamics have passed. We have added these details to the revised manuscript (lines 576-577, and Supplementary Methods 1, section 2).

--- Appendix S1 = At page 2 it is stated that “cell update consists of the following ordered processes...” Does it mean state variables are updated in an asynchronous way (i.e. the new values are not stored until all individuals have executed the process, but updated in a sequential way within each time step – e.g. this can have an effect on when trophic interactions occur; see also Appendix S1 at page 4)?

Yes, state variables are updated asynchronously to avoid unrealistic events or phenomena. For example, if a given individual in a given time step moves to a new cell, it would be erroneous to assume that another individual in an adjacent cell could encounter the former individual in the cell where it was located before. We have now clarified this.

--- Is there an effect of satiety for feeding interactions? I could not find this aspect mentioned in the Appendix S1. Please, add one sentence on satiety to say whether it is considered or not.

We do not consider satiety. We assume consumers always take resources. This is a realistic assumption as all individuals are subject to energy expenditure in each time step, which means that they will always need to gather resources to stay alive. We now have a sentence to specify this.

--- The way the parameter space was validated (i.e. “...a set of parameter values were selected that produced realistic community patterns and stable dynamics.”) does not seem properly documented. Do the authors also refer to the sensitivity analysis of Appendix S2? I expect some more emphasis on this choice in the main body of the manuscript, especially considering that default values of model parameters (Table 1, which should be Table S1) were not obtained from real empirical data. Also, what does exactly mean that “realistic community patterns and stable dynamics” were obtained? This is too vague and a more quantitative justification is needed.

The reviewer is right in asserting that model parameters have not been obtained empirically, and the way in which the values were obtained needs to be properly (i.e., more quantitatively) justified. Parameter values come from published results of this model (Lurgi et al, 2016) that show that the resulting simulated communities display patterns similar to those observed in real communities. Some of these quantitative patterns include log-normal rank-abundance distributions and exponential degree distributions in the food web. We have now added figures presenting these patterns in the supplementary material (Supplementary figures S6, S7). Some parameter values are based on ecological realism, e.g. assimilation rate is higher for plant biomass than animal biomass (herb efficiency > efficiency trans). Despite this, sensitivity analysis shows that results are robust to variations in parameter values (Supplementary methods 3). We now include more material on parameter selection in the main text, and on the limitations paragraph (lines 411-418, 639-648, Supplementary Methods 1, section 1.2). We have also relabelled table 1 to table S1.

--- Appendix S1, at page 4 = Why the choice of offspring placed in a range of 3 cells of distance? Which is the distance in meters? Are there literature references to support such choice? How is the choice of when to release the seed taken by the herbivore in case of mutualistic interactions (i.e. in terms of number of steps)? Are these choices mediated by probabilities? Were these probabilities estimated using empirical data? If yes, literature references should be indicated. More in general, the correspondence of cells with spatial scale (one cell = 1 square meter?) needs to be supplied.

The choice of three cells of distance for offspring is arbitrary, consistent with other model choices, and based on the fact that offspring of many species tend to be close to their parents range during the early stages of life (they can move afterwards, and this is considered in the model). Sensitivity analysis confirms that using other values yields similar results. We now include this justification in the Supplementary Methods 1.

We do not specifically assign any distance units or spatial scale on purpose. This is because our goal is not to represent a particular terrestrial community or specific ecosystem, but to investigate the response of a standard / ideal community to habitat loss. Defining distance units more specifically would be highly dependent on the system studied, and would therefore be meaningless in this study (see also comment above related with the spatial scale). We have specified this in the revised manuscript (lines 466-470).

Seed release by herbivores is a decaying function of time and depends on a given probability. We have re-written this section in the supplementary methods to make this clearer.

We have included more material on parameter selection and probabilities in the main text, and added material in the discussion (see response to previous comment).

--- Appendix S1, last page = My understanding is that at each simulation step of the random habitat loss one cell is destroyed. Is this correct? Does simulation runs last well beyond the moment the last cell is removed? If yes, for how long? Why (is this related to the need of attaining a stable condition after the transient phase)? Does the same condition (i.e. one cell removed per time step) apply to the case of contiguous habitat loss? If not, can these differences in cell removal scenarios between the two types of habitat loss affect the results?

Firstly, undisturbed communities with stable dynamics are generated. Then habitat is removed in successive steps (% of habitat loss) and we let disturbed communities at each % of habitat loss to evolve and pass a transient phase. Subsequently, the model computes the metrics for that given % of habitat loss. The same procedure is used for all types of habitat loss. We are more explicit about this in the revised manuscript.

--- Last line of Appendix S1 = Sometimes the information provided is technically correct but does not help non-experts to understand the reasons of the choices taken (e.g. when talking about toroidal boundaries that, I guess, were considered to avoid edge effect during simulations). I think that more explicit and intuitive explanations should be given, especially if the goal is being of appeal for a wide audience of ecologists.

The reviewer is right and we have included less technical explanations when possible.

The second aspect that should be improved in the manuscript relates to the structure of the introduction. There are poorly defined concepts and the adoption of some definitions strikes with mainstream classifications in ecology. Sometimes, the manuscript is biased towards the

modelling perspective and risks to be detached from terms and definitions that are common in experimental and theoretical ecology. The contribution of the authors is valuable but they should not degrade ecology to modelling. Also in discussions and conclusions I would like to read more considerations about the ecological value of their findings, rather than general statements that are poorly coordinated with the rest of the manuscript (e.g. at L311-312 they first presented possible value of the IBM to explore the role of global changes in shape local diversity – this seems to be poorly coordinated with the rest of the manuscript and could be removed).

We have removed technical jargon when possible, and added elements of discussion earlier in the introduction to coordinate them better. We have also followed the suggestions made by the reviewer below.

I provide below a list of points that should be taken into account to improve the introduction:
--- L67-69 = I suggest the sentence should be rewritten. I do not understand how “extinction thresholds are higher when habitat loss is spatially-correlated” (compared to random loss) “because spatially-correlated destruction leaves larger areas of pristine habitat intact”. If larger areas of pristine habitat are intact I would expect less extinction. Maybe the confusion is related to the use of the terms “extinction thresholds”. Anyway, clear statement of the causative mechanism is required in the text.

We have re-written this sentence.

--- L78-80 = I was a bit puzzled by the definition of food web proposed here. Food webs depict trophic interactions (no matters whether they are plant-herbivore or predator-prey); I would replace the terms “food webs” with “ecological networks” or “network studies of ecological communities” to better accommodate also the plant-pollinator interactions.

Done.

--- L88-92 = The concept of stability is central for this manuscript. While I prefer a mathematical definition of stability, which cannot be applied to this study (as it requires the presence of ODEs; see Neutel et al. 2002), I understand the need of the authors to rely on different ways to quantify stability. Nevertheless, they should be more explicit since the beginning about the meaning of “stability”. First, which are the other aspects of stability that can be affected by habitat loss (L89)? Mention these alternative aspects before citations. Second, the various definitions of stability are correlated. For example, more extinction events often result in higher levels of spatial heterogeneity and change the structure and functioning of the food web. This is the case of meso-predator release due to top-predator extinction from local patches, which is triggered by habitat fragmentation in coastal southern California (Crooks and Soulé 1999). Also, changes in food web architecture (e.g. reduction of interactions) can sharpen the risks of secondary extinctions following primary extinction events (Allesina and Bodini 2004). These aspects should be made explicit in the text.

We have addressed the reviewer’s comments by specifying other aspects of stability that can be potentially affected by habitat loss (lines 91-92). We have also make explicit mentions to the correlative nature of stability metrics (lines 93-95, 361-367), and cited Allesina and Bodini (2004) appropriately (line 95).

--- L109-113 = Which is the way used to quantify stability (e.g. stability in the food web structure, in the number of species or related spatial and temporal heterogeneity)?

We have now clarified this aspect (lines 118-121).

--- L114-118 = The meaning of “types of habitat loss” (intended as dynamics following either

random of continuous patterns) gets clear only at the end of the introduction. I am convinced the reader would benefit of an earlier clarification of this concept, which is central for the manuscript.

We have clarified this earlier in the revised manuscript (lines 30, 65-77)

Clear definition of interaction strength should be stated as there are many strategies for quantifying the strength of interactions (see Berlow et al. 2004). The definition of strength is essential because it allows establishing an indirect causative link between habitat loss and stability (especially in the case of contiguous habitat loss for which average interaction strength increases and communities become more unstable). How did the authors convert the various individual-level interactions in the different cells in terms of interaction strength? Maybe I lost the description of this aspect in the text, but for sure more emphasis and explicit description should be dedicated to it. How were the interactions deduced for the taxa starting from individual-level information? Was the strength defined in terms of frequency of interactions between pairs of species? If yes, was the frequency weighted in terms of energy/biomass flowing from resources to consumers? These aspects have consequences on the definition of stability (L407-425) but also influence the quantitative version of many network indices (L395-405 and Table 1 in the manuscript).

Interaction strength was quantified as the number of predation (or pollination, as an example of mutualistic interaction) events that happened between individuals of the species involved in the links (i.e. the number of individuals of the prey species consumed by individuals of the predator species) divided by the product of both abundances. This quantifies the per capita effect of a predator species over the population of its prey, and is analogous to Paine's index and Lotka-Volterra interaction coefficients. We have corrected the definition of mean interaction strength (there was a mistake in table 1) and clarified this aspect in the revised manuscript.

The manuscript would benefit of a paragraph in the discussion where possible limits of the modelling framework adopted should be clearly summarized. Such limits refer to some purely methodological aspects listed above (e.g. use of parameters not inferred from empirical or experimental data; consideration of unrealistic conditions to test the role of habitat fragmentation only respect to species richness – see L283-286) but should also honestly present the ecological limits of their application (e.g. the IBM is adapt for the study of terrestrial systems but not suitable for aquatic food webs).

We have now added a paragraph describing the limitations of this study (lines 385-421), including those highlighted by the reviewer.

Minor comments

--- L36-37 = How can the authors state their results are generalizable to communities of multiple species? I pointed out possible issues in the application to aquatic systems (in the general comments), which already represent a clear limit to generalization. Also, the lack of biological evidence for most of the parameters adopted in the IBM consists of a limit to talk about generalization. I recommend removing this sentence and clearly state the application is particularly suitable to terrestrial food webs.

We now explicitly state that our model applies to terrestrial communities in the introduction and discussion sections (lines 29-30, 273-275, 398-403, 421, 466-468).

--- L37-39 = Instead of writing that the causal chains (operating behind responses to two schemes of habitat loss) have been revealed, I would appreciate the explicit description of these

two mechanisms. One is the increase of interaction strength that leads to instability (in case of contiguous habitat loss) and another is the impact on diversity and network structure that depend on habitat fragmentation.

Done.

--- L38 = While reading the abstract and the introduction, it was not fully clear to me the meaning of “types of habitat loss”. The reader could have immediately a better idea about one key aspect of the work if the authors would state in the abstract that the two types of habitat loss investigated are random and contiguous. They should use more these terms in the introduction (e.g. see L64-67) as well, since in the version I reviewed they got clear only starting from the methods section. Moreover, the fact the authors refer to the type of habitat loss with different terms might increase confusion (e.g. L64 = “how destruction takes place”; L74 “nature of habitat loss”; L358: “two habitat loss scenarios”).

Done. We have used a similar terminology of the types of habitat loss as much as possible. In some cases, though, we have kept synonymic terms to avoid redundancy.

--- L47 = Replace “especial” with “particular”

Done.

--- L58 = Remove “s”: “empirical data on insect food-webs show that...”

Done.

--- In the whole manuscript = I suggest using the hyphen when writing “individual-based model”.

Done.

--- L169-171 = This part (with citation) does not belong to the results; it should be moved to discussion (eventually).

We have rephrased this part providing a result (and its corresponding plot) of this study.

--- RATP is mentioned for the first time in the text at L186 and its definition is provided in Table 1 only. I suggest including the meaning of RATP the first time is mentioned in the text (and to include it in the text, with the full name of such index).

Done.

--- L217-219 = I do not see how the authors can state their findings (i.e. the link between habitat area, interactions strength and stability) are generalizable to communities of multiple species and interaction types. Rather, I highlighted how their application has validity in a specific type of systems (i.e. terrestrial communities) and conclusions cannot be drawn for other interaction types without further analysis (e.g. social interactions), since the inclusion of other interaction types can escape the logic defined by the specific functions codified for the IBM presented here.

Changed. See also response to a previous comment by the reviewer.

--- L390-393 = Did the authors consider quantifying the diversity of the network structure?

We are not sure we understand the reviewer's comment. We have defined diversity metrics as those depicting aspects related with the number of species and their abundances. The diversity of network structure has been quantified using network metrics commonly employed in food-web studies: for example, number of links, which quantifies link richness; connectance, which quantifies the fraction of realised interactions; vulnerability and generality, which quantify the diversity of interaction partners that species in lower and upper trophic levels have, among others.

--- L414-417 = I would be careful with these statements. Asynchrony in the abundance/biomass fluctuations across trophic groups can also contribute to stabilize ecosystem functioning (e.g. keeping constant biomass production in spite of seasonal fluctuations). Here fluctuations appear to correspond with a negative property (i.e. instability), but such consideration should be attenuated as often fluctuations are the key to maintain constant ecosystem functioning and the provision of services. Some words should be spent by the authors to clarify this aspect.

We have now clarified this in the revised manuscript (lines 595-600).

--- L475 = "plant-pollinator" (remove an extra hyphen).

Done.

--- There are too many references, an issue that often translates in blurred messages. One possible way to reduce them is by removing some redundant ones (e.g. at line 84, suggestions 21-27: remove those not used anymore in the text, if any).

We have removed redundant references when possible. However, the comments made by the reviewers also forces us to add references that were not included in the original manuscript. Thus, new references are inevitably provided in the revised manuscript. Despite this, we have been able to maintain the same number of references as in the original manuscript.

--- Caption of Figure 1 (line 6) = Do the authors mean that normalization was obtained by setting the maximum value of each row to 1? If yes, say it explicitly.

Done.

--- Caption of Figure 2 (line 1) = "Interaction strength..." (remove "s").

Done.

--- Figure 2 = Which is the reason for the differences in the values plotted in panels A and B when habitat loss is 0? I thought this initial point should show equivalent distribution of values for IS and CV_pop (irrespectively of the habitat loss scenario) when no habitat loss occurred yet (i.e. x-axis value = 0). I also suggest using the same range over y-axis for both panels A and B, especially for CV_pop.

It is correct that the values in panels A and B at HL=0 do represent the same underlying distribution regardless of habitat loss scenario, but each panel shows 25 different samples from that underlying distribution corresponding to 25 distinct stochastic simulation runs. We have amended the figure (now figure 3) and subsequent analysis in the revised manuscript to use the same set of 25 simulation runs at HL=0 in all habitat loss scenarios. We now use the same y-axis ranges across panels for clarity as suggested.

--- Table 1 = There are a series of mistakes (or, at least, improper wordings): (1) the terms b.k

and b.. refer to biomasses in case of both generality and vulnerability (these are properly defined in Lurgi et al. 2015, but here the definitions were modified and lost their correct meaning); (2) in nestedness, you should remove the repetition of “of” under the metric column; (3) I am not sure b_{ij} is the number of interactions, as written in the definition of Mean IS (I would expect this to be related to actual biomass flowing from prey i to predator j – this might also help to address my concerns related to the definition of interaction strength); (4) usually CV is expressed as %; (5) not clear the rational reason for focusing on the last 200 simulation steps to calculate CV population and CV range (it should be explicitly mentioned, at least in the text).

(1-4) Done; (5) The main reason is to avoid misleading results due to transient dynamics. We considered the last 200 time steps to be stable and a large enough number to estimate statistics. We have now added these details (lines 588-589, supplementary methods 1).

--- Appendix S1 needs some text editing (e.g. include only the year “2015” in parenthesis for the citation at page 1, line 2; adjust citation format in the caption of Figure 1 – please double check the whole document to fix this type of issues). Also, in the footnote of the first page it should be “...is also provided in the supporting...”

We have improved Appendix S1, and removed the footnote.

--- Appendix S1 = At page 6 the reference should be to Appendix S2 and the authors should remove the following words: “Throughout the thesis”.

Done.

--- Appendix S2 = The authors stated the criteria that led at the selection of the variables used for the SEM (e.g. RATP, Links, IS). I liked the clear description but do not understand why the number of individuals was excluded (as it shares all of the features displayed by the variables considered in the SEM). An explanation for the exclusion of the variable “number of individuals” is needed.

The number of individuals was excluded from the analysis because it changed exactly in the same way across habitat loss scenarios. We now include this explanation.

--- Appendix S2 = The last paragraph refers to MAI ratio that was never defined in the manuscript (it is presented in Lurgi et al. 2015); also, this last paragraph is poorly coordinated with the rest of the Appendix S2. Moreover, which is the meaning of ACV?

We have corrected these issues in the revised manuscript.

--- Some adjustments are needed in Table 1 of Appendix S2 (it should be Table S2): (1) provide references for the argument of HL – Links; (2) rewrite HL to IS as “...interaction strengths is offered by Tylianakis et al. (2008) and Hagen et al. (2012).”; (3) Links to RATP, the citation should be at the end of the definition; (4) Links to CV population and CV range, rewrite as: “...May’s seminal work (1972) is that...”; (5) what stated in the motivation of the link RATP to IS can also be not completely true if we consider the non-random and non-homogeneous distribution of strong and weak links in food webs (see Neutel et al. 2002) – I would be careful with the statement provided and include my thoughts; (6) in the definition of the link from IS to CV range rewrite as “perhaps”; (7) in the link from CV range to CV population remove the repetition for the word “types”.

We have made all the adjustments suggested by the reviewer.

--- Appendix S3 = There is a weird self-citation to Appendix S3, and the numbering of Figure and Table should consider the name of the Appendix (e.g. Figure S3 and Table S3). Also, in a couple of cases the indication ± 20 misses the symbol “%” and there is a wrong reference to Appendix S1 for SEM (it should be Appendix S2).

Corrected.

--- Supplementary Figures should be renumbered (this applies to all appendices, to avoid having many Figures 1 for example) and the resolution improved.

Done.

References

- Allesina, S. and Bodini, A., 2004. Who dominates whom in the ecosystem? Energy flow bottlenecks and cascading extinctions. *Journal of Theoretical Biology* 230:351-358.
- Berlow, E.L., Neutel, A.M., Cohen, J.E., De Ruiter, P.C., Ebenman, B.O., Emmerson, M., Fox, J.W., Jansen, V.A., Iwan Jones, J., Kokkoris, G.D. and Logofet, D.O., 2004. Interaction strengths in food webs: issues and opportunities. *Journal of Animal Ecology* 73:585-598.
- Black, A.J. and McKane, A.J., 2012. Stochastic formulation of ecological models and their applications. *Trends in Ecology & Evolution* 27:337-345.
- Bolnick, D.I., Amarasekare, P., Araújo, M.S., Bürger, R., Levine, J.M., Novak, M., Rudolf, V.H., Schreiber, S.J., Urban, M.C. and Vasseur, D.A., 2011. Why intraspecific trait variation matters in community ecology. *Trends in Ecology & Evolution* 26:183-192.
- Crooks, K.R. and Soulé, M.E., 1999. Mesopredator release and avifaunal extinctions in a fragmented system. *Nature* 400:563-566.
- García-Callejas, et al. 2017. Spatial trophic cascades in communities connected by dispersal and foraging. *bioRxiv* 469486.
- Giacomini, H.C., De Marco Jr, P. and Petre Jr, M., 2009. Exploring community assembly through an individual-based model for trophic interactions. *Ecological Modelling* 220:23-39.
- Jacob, S. et al. 2018. Habitat choice meets thermal specialization: Competition with specialists may drive suboptimal habitat preferences in generalists. *PNAS* 115 (47) 11988-11993
- Lurgi, M., Montoya, D. and Montoya, J.M. 2016. The effects of space and diversity of interaction types on the stability of complex ecological networks. *Theoretical Ecology* 9:3-13.
- Neutel, A.M., Heesterbeek, J.A. and de Ruiter, P.C., 2002. Stability in real food webs: weak links in long loops. *Science* 296:1120-1123.
- Van Tussenbroek, B.I., Villamil, N., Márquez-Guzmán, J., Wong, R., Monroy-Velázquez, L.V. and Solis-Weiss, V., 2016. Experimental evidence of pollination in marine flowers by invertebrate fauna. *Nature Communications* 7:12980.

Reviewer #1 (Remarks to the Author):

Review of revised "Routes to instability under habitat loss: an investigation of multitrophic communities" by McWilliams et al.

I appreciate the authors' responses to my comments. They have addressed my concerns satisfactorily. The revised version of the manuscript is notably improved. I am supportive of publication in Nature Communications.

I offer three further suggestions regarding the point about the fraction of mutualistic interactions: (i) in addition to the average fraction of mutualistic interactions, 23%, please add the standard deviation; (ii) provide some empirical context for the expected fraction of mutualistic interactions in an ecological community (i.e., is 23% too low, too high, about right for a real community?); and (iii) consider the possibility of including mutualistic interactions in a configuration that is more typical of mutualistic networks (e.g., binary nestedness pattern) rather than trophic networks (i.e., derived from the niche model).

I sign my review: Phillip P.A. Staniczenko

Reviewer #3 (Remarks to the Author):

General comments

The Authors did an excellent job in addressing all of the questions I raised during the previous round. Changes in both the manuscript and Supplementary Material files are appropriate and significantly increased the quality of the manuscript.

The authors clearly motivated the reason behind the choice of an individual-based model (IBM) approach (vs. ordinary differential equations) and performed new sensitivity analysis to show the marginal effect of migration rates on model outcomes. Moreover, they provided many details concerning the IBM (e.g. state variables are updated asynchronously, satiety is not considered and consumers always take resources when available, the same probability for immigration is used for all individuals, and all non-destroyed cells in the landscape grid exhibit the same habitat quality); such details are essential to allow the readers reproducing the model and ensure the required transparency to fully evaluate the premises on which model outcomes depend.

The Introduction and other parts of the text (e.g. Discussion) were modified to provide clear descriptions about the two extreme types of habitat loss (random and contiguous) considered in the manuscript, and also to indicate the meaning of stability (i.e. temporal and spatial variability in population abundances). Moreover, I appreciated the paragraph of the Discussion (lines 385-421) where the limits of the IBM (and possible ways for its improvement) are presented.

There are still a few minor points (mainly related to text editing and typos) that the Authors should take into account prior considering the manuscript for publications (see the list of minor comments below).

Minor comments

Line 76 – “...thus on...”

Lines 118-121 – this part of the text describes how temporal variability in population abundances (CV population) and the variability in species’ range areas (CV range) represent two response variables for measuring stability. Analogous approach (i.e. based on CV) was previously adopted to quantify the heterogeneity of spatial distribution of species and individuals simulated by an IBM (Scotti et al. 2013). I suggest including this reference to the text.

Scotti, M., Ciocchetta, F., & Jordán, F. (2013). Social and landscape effects on food webs: a multi-level network simulation model. *Journal of Complex Networks*, 1(2), 160-182.

Line 132 – Remove the hyphens: “...HL (i.e. random and contiguous), which...”

I suggest doing the same at lines 135 and 137

Line 162 – “Sensitivity analysis shows that...”

Line 215 – “color scale”

Lines 255-257 – The use of the letters to indicate the charts 5B and 5D seems to be wrong. Contiguous loss is in Fig. 5D while random loss is in Fig. 5B

Line 269 – “Our analysis of three scenarios of HL suggests...”

Line 427 – “HL not only reduces...”

Lines 441-442 – Check verb tense (please, do it in the whole manuscript): “We used an individual-based...and extend it to...” Past vs. present in this sentence

Lines 509-511 – This sentence could also be moved to Results. In the Methods the Authors could state that the dependence of model outcomes on immigration rates is tested with sensitivity analysis, which serves to quantify the impact of HL on IS and variability of population abundances

Lines 518-519 – I suggest rephrasing the following sentence: “Basal species (plants) may only occupy the inhabitant space, whilst all other species may occupy either or both spaces.” Does it mean that a basal species cannot move (and thus can only occupy the cell where it was placed at the beginning of simulation) while consumers are free to move on the grid? Which are “both spaces”? Are these spaces the “inhabitant” and “visitor” ones? What does it mean that non-basal species can occupy both?

Line 554 – “We expected that the two extreme HL types have contrasting effects on...”

The definition of Nestedness in Table 1 must be fixed. In the present version it is not correct.

Line 823 – “p-value > 0.05” (use the decimal point)

Lines 832-834 – I suggest revising the text of these last two sentences in the caption of Figure 2. I have the feeling some parts of the text are missing

Figure 2 on RDA analysis is not visualized in the correct way as the chart looks empty (no points on the 2D space that only includes the legend in the upper right corner)

Line 869 – Add that green color refers to destroyed cells under the intermediate HL scenario

Supplementary Methods 2, Page 2, Line 25 – “On each iteration, the SEM was refitted and one link removed...”

Supplementary Methods 2, Table S1, row “HL, IS” – in the column “Argument for causal link” the words “interaction strengths” are repeated twice

Supplementary Methods 2, Table S1, row “Links, IS” – in the column “Argument for causal link”: “...weaker, which to reduce the mean IS.” Please, rephrase

Supplementary Methods 2, Table S1, row “Links, CV population and CV range” – in the column “Argument for causal link”: “...is likely to increase...” or “...is likely increasing...”

Response to reviewer's

Reviewer #1 (Remarks to the Author):

Review of revised "Routes to instability under habitat loss: an investigation of multitrophic communities" by McWilliams et al.

I appreciate the authors' responses to my comments. They have addressed my concerns satisfactorily. The revised version of the manuscript is notably improved. I am supportive of publication in Nature Communications.

We thank Phillip P. A. Staniczenko for his insightful comments on this manuscript, which have substantially improved the manuscript.

I offer three further suggestions regarding the point about the fraction of mutualistic interactions: (i) in addition to the average fraction of mutualistic interactions, 23%, please add the standard deviation; (ii) provide some empirical context for the expected fraction of mutualistic interactions in an ecological community (i.e., is 23% too low, too high, about right for a real community?); and (iii) consider the possibility of including mutualistic interactions in a configuration that is more typical of mutualistic networks (e.g., binary nestedness pattern) rather than trophic networks (i.e., derived from the niche model).

(i) In the revised manuscript, we add the standard deviation of the fraction of mutualism.

(ii) Providing an empirical context for the fraction of mutualism is not straightforward as most empirical network studies sample only a part of the meta-web, e.g. either two trophic levels or single types of interactions. Besides, it is likely that the fraction of mutualism in real communities will be habitat-dependent. Despite this, some examples of empirical meta-webs do exist. For example, in Montoya et al (Nature Communication 2015), various types of species and interactions in a saltmarsh ecosystem were sampled in two consecutive years. The average cross-year fraction of mutualistic interactions in this study is 24%, with a standard deviation of 15.5, which is pretty similar to what our simulated communities reflect. In Pocock et al (Science 2012), mutualist species (flower visitors and granivorous animals) in a farmland landscape represented around a 22% of the total number of species and 58% of interactions (although this number may be lower as not all seeds are dispersed by mutualists), which included plants and 11 groups of animals: those feeding on plants (butterflies and other flower visitors, aphids, seed feeding insects, and granivorous birds and mammals) and their 'dependants' (parasitoids and ectoparasites). We have added this information in the revised manuscript.

(iii) We appreciate the third point made by the reviewer. However, we wanted to be consistent when generating the full network of species interactions. The niche

model describes trophic niche occupancy between consumers and resources along a resource axis and successfully generates network structures that approximate well the central tendencies and the variability of a number of food web properties (Williams and Martinez 2000; Dunne et al. 2002; Stouffer et al. 2005). Because it arranges consumers and resources along a resource axis, the niche model can be applied to other types of consumer-resource interactions, including mutualistic interactions such as pollination (see Holland et al. 2005; Holland and De Angelis 2009; Holland 2015). Besides, the initial values of nestedness lie within the empirical range in mutualistic webs (15-30, Supplementary Figure S2; Thebault & Fontaine 2010). The revised manuscript includes a justification of this.

References

- Dunne JA, Williams RJ, Martinez ND (2002) Food-web structure and network theory: the role of connectance and size. *PNAS* 99:12913– 12916
- Holland, J. N. et al. 2005. Mutualisms as consumer-resource interactions. Pages 17-34. *Ecology of Predator-Prey Interactions*. Oxford University Press, Oxford, UK.
- Holland, J. N. & De Angelis, D. L. 2009. Consumer-resource theory predicts dynamic transitions between outcomes of interspecific interactions. *Ecology Letters* 12, 1357-1366.
- Holland, J. N. 2015. Population ecology of mutualism. Pages 133–158 in J. Bronstein, editor. *Mutualism*. Oxford University Press, Oxford, UK.
- Stouffer DB, Camacho J, Guimera R, Ng C, Nunes Amaral L (2005) Quantitative patterns in the structure of model and empirical food webs. *Ecology* 86:1301–1311
- Thebault E, Fontaine C (2010) Stability of ecological communities and the architecture of mutualistic and trophic networks. *Science* 329: 853–856
- Williams RJ, Martinez ND (2000) Simple rules yield complex food webs. *Nature* 404:180–183

I sign my review: Phillip P.A. Staniczenko

Reviewer #3 (Remarks to the Author):

General comments

The Authors did an excellent job in addressing all of the questions I raised during the previous round. Changes in both the manuscript and Supplementary Material files are appropriate and significantly increased the quality of the manuscript.

The authors clearly motivated the reason behind the choice of an individual-based model (IBM) approach (vs. ordinary differential equations) and performed new sensitivity analysis to show the marginal effect of migration rates on model outcomes. Moreover, they provided many details concerning the IBM (e.g. state variables are

updated asynchronously, satiety is not considered and consumers always take resources when available, the same probability for immigration is used for all individuals, and all non-destroyed cells in the landscape grid exhibit the same habitat quality); such details are essential to allow the readers reproducing the model and ensure the required transparency to fully evaluate the premises on which model outcomes depend.

The Introduction and other parts of the text (e.g. Discussion) were modified to provide clear descriptions about the two extreme types of habitat loss (random and contiguous) considered in the manuscript, and also to indicate the meaning of stability (i.e. temporal and spatial variability in population abundances). Moreover, I appreciated the paragraph of the Discussion (lines 385-421) where the limits of the IBM (and possible ways for its improvement) are presented.

We than the reviewer for his/her comments on the manuscript, which have increase the quality of the manuscript.

There are still a few minor points (mainly related to text editing and typos) that the Authors should take into account prior considering the manuscript for publications (see the list of minor comments below).

Minor comments

Line 76 – “...thus on...”

Done.

Lines 118-121 – this part of the text describes how temporal variability in population abundances (CV population) and the variability in species’ range areas (CV range) represent two response variables for measuring stability. Analogous approach (i.e. based on CV) was previously adopted to quantify the heterogeneity of spatial distribution of species and individuals simulated by an IBM (Scotti et al. 2013). I suggest including this reference to the text.

Scotti, M., Ciocchetta, F., & Jordán, F. (2013). Social and landscape effects on food webs: a multi-level network simulation model. *Journal of Complex Networks*, 1(2), 160-182.

Done.

Line 132 – Remove the hyphens: “...HL (i.e. random and contiguous), which...”
I suggest doing the same at lines 135 and 137

Done.

Line 162 – “Sensitivity analysis shows that...”

The paragraph has been removed, but we have made the appropriate changes in other parts of the manuscript according to the reviewer's suggestion.

Line 215 – “color scale”

Done.

Lines 255-257 – The use of the letters to indicate the charts 5B and 5D seems to be wrong. Contiguous loss is in Fig. 5D while random loss is in Fig. 5B

We have corrected this in the manuscript.

Line 269 – “Our analysis of three scenarios of HL suggests...”

Done.

Line 427 – “HL not only reduces...”

Done.

Lines 441-442 – Check verb tense (please, do it in the whole manuscript): “We used an individual-based...and extend it to...” Past vs. present in this sentence

Done.

Lines 509-511 – This sentence could also be moved to Results. In the Methods the Authors could state that the dependence of model outcomes on immigration rates is tested with sensitivity analysis, which serves to quantify the impact of HL on IS and variability of population abundances

Done.

Lines 518-519 – I suggest rephrasing the following sentence: “Basal species (plants) may only occupy the inhabitant space, whilst all other species may occupy either or both spaces.” Does it mean that a basal species cannot move (and thus can only occupy the cell where it was placed at the beginning of simulation) while consumers are free to move on the grid? Which are “both spaces”? Are these spaces the “inhabitant” and “visitor” ones? What does it mean that non-basal species can occupy both?

We have rephrased this sentence to improve clarity.

Line 554 – “We expected that the two extreme HL types have contrasting effects on...”

Done.

The definition of Nestedness in Table 1 must be fixed. In the present version it is not correct.

Done.

Line 823 – “p-value > 0.05” (use the decimal point)

Done.

Lines 832-834 – I suggest revising the text of these last two sentences in the caption of Figure 2. I have the feeling some parts of the text are missing

We have re-worded the caption.

Figure 2 on RDA analysis is not visualized in the correct way as the chart looks empty (no points on the 2D space that only includes the legend in the upper right corner)

We provide a corrected figure.

Line 869 – Add that green color refers to destroyed cells under the intermediate HL scenario

Done.

Supplementary Methods 2, Page 2, Line 25 – “On each iteration, the SEM was refitted and one link removed...”

Done.

Supplementary Methods 2, Table S1, row “HL, IS” – in the column “Argument for causal link” the words “interaction strengths” are repeated twice

Done.

Supplementary Methods 2, Table S1, row “Links, IS” – in the column “Argument for causal link”: “...weaker, which to reduce the mean IS.” Please, rephrase

Done.

Supplementary Methods 2, Table S1, row “Links, CV population and CV range” – in the column “Argument for causal link”: “...is likely to increase...” or “...is likely increasing...”

Done.